# Boosting hydrogel conductivity via water-dispersible conducting polymers for injectable bioelectronics

Hossein Montazerian [1,2,3,4,14], Elham Davoodi[3,5,14], Canran Wang[5],
Farnaz Lorestani[6], Jiahong Li[5], Reihaneh Haghniaz[4], Rohan R. Sampath [7],
Neda Mohaghegh[4], Safoora Khosravi[4,8], Fatemeh Zehtabi[4], Yichao Zhao[1],
Negar Hosseinzadeh[4], Tianhan Liu [7], Tzung K. Hsiai [2],
Alireza Hassani Najafabadi [4] ✉, Robert Langer [1,9,10,11,12],
Daniel G. Anderson [1,9,10,11,12], Paul S. Weiss[2,7,13] ✉, Ali Khademhosseini [4] ✉ &
Wei Gao [5] ✉

Bioelectronic devices hold transformative potential for healthcare diagnostics and therapeutics. Yet, traditional electronic implants often require invasive surgeries and are mechanically incompatible with biological tissues. Injectable hydrogel bioelectronics offer a minimally invasive alternative that interfaces with soft tissue seamlessly. A major challenge is the low conductivity of bioelectronic systems, stemming from poor dispersibility of conductive additives in hydrogel mixtures. We address this issue by engineering doping conditions with hydrophilic biomacromolecules, enhancing the dispersibility of conductive polymers in aqueous systems. This approach achieves a 5-fold increase in dispersibility and a 20-fold boost in conductivity compared to conventional methods. The resulting conductive polymers are molecularly and in vivo degradable, making them suitable for transient bioelectronics applications. These additives are compatible with various hydrogel systems, such as alginate, forming ionically cross-linkable conductive inks for 3D-printed wearable electronics toward high-performance physiological monitoring. Furthermore, integrating conductive fillers with gelatin-based bioadhesive hydrogels substantially enhances conductivity for injectable sealants, achieving 250% greater sensitivity in pH sensing for chronic wound monitoring. Our findings indicate that hydrophilic dopants effectively tailor conducting polymers for hydrogel fillers, enhancing their biodegradability and expanding applications in transient implantable biomonitoring.

Bioelectronic devices have transformed the landscape of medical diagnosis and treatment due to their immense potential in sensing biosignals and stimulating impaired tissues[1,2]. However, interfacing these devices with internal organs often requires invasive surgeries, which, coupled with mechanical mismatches with the tissue microenvironment, lead to major complications in their long-term performance. These complications stem primarily from fibrosis, poor integration, and damage to the surrounding native tissue. Soft injectable bioelectronics[3,4] are emerging as a promising solution, enabling favorable tissue interfacing through

minimally invasive approaches[5–7], such as delivery via needles and catheters[8–10].

Hydrogels have shown excellent versatility for seamless integration with injectable platforms, driving the demand for conductive hydrogels as injectable bioelectronics[11]. Hydrogel bioelectronic devices are created by the incorporation of conductive additives in hydrophilic polymer networks[12]. These networks can be engineered further to introduce various functionalities, including tissue regenerative effects, stimuli-responsiveness, bioadhesion, and more[13–16]. Traditional conductive materials like metals and carbon-based additives, despite their high conductivity, pose risks of immunogenicity and cytotoxicity[17,18]. In contrast, conductive polymers, particularly poly(3,4-ethylenedioxythiophene):poly(styrene sulfonate) (PEDOT:PSS), offer biocompatibility and tunable chemistry but face challenges with aggregation and poor percolation networks, resulting in low conductivity[19].

Efforts to enhance hydrogel conductivity with PEDOT have reported impressive results[20], yet these methods often involve processes such as drying steps and cytotoxic phase separation triggers[11,21–26], making them unsuitable for injectable bioelectronics. Consequently, much of the literature has used PEDOT:PSS as fillers to impart conductivity to hydrogel platforms for minimally invasive and

3D printing applications[27–30]. The primary limitation of PEDOT hydrogel composites is their poor dispersibility in aqueous systems due to the aggregation driven by hydrophobic groups in the PSS backbone, which restricts their conductivity.

To address these challenges and to achieve high-performance injectable bioelectronics, we introduce a strategy to boost PEDOT hydrogel conductivity by using naturally derived hydrophilic dopants[27,31–35] instead of PSS (Fig. 1a). We hypothesize that hydrophobic polystyrene backbone of PSS contributes to the poor dispersibility of dry PEDOT:PSS. Thus, we chose alginate, with its rich content of polar groups, as a hydrophilic backbone and modified it with sulfonates (sulfonated alginate, AlgS) that serve as a doping agent in lieu of PSS in the PEDOT polymerization process. The enhanced hydrophilicity allows the freeze-dried PEDOT:AlgS to be re-dispersed in hydrogels at concentrations approximately five times higher than PEDOT:PSS, enabling an order of magnitude improvement in achievable conductivity thresholds in hydrogels (Fig. 1b). Additionally, this approach provides molecular-level and in vivo degradability for transient bioelectronics applications.

We demonstrate the utility of PEDOT:AlgS in alginate matrices to develop ionically cross-linkable conductive inks for 3D printing of soft bioelectrodes where PEDOT:AlgS enables highly sensitive detection of

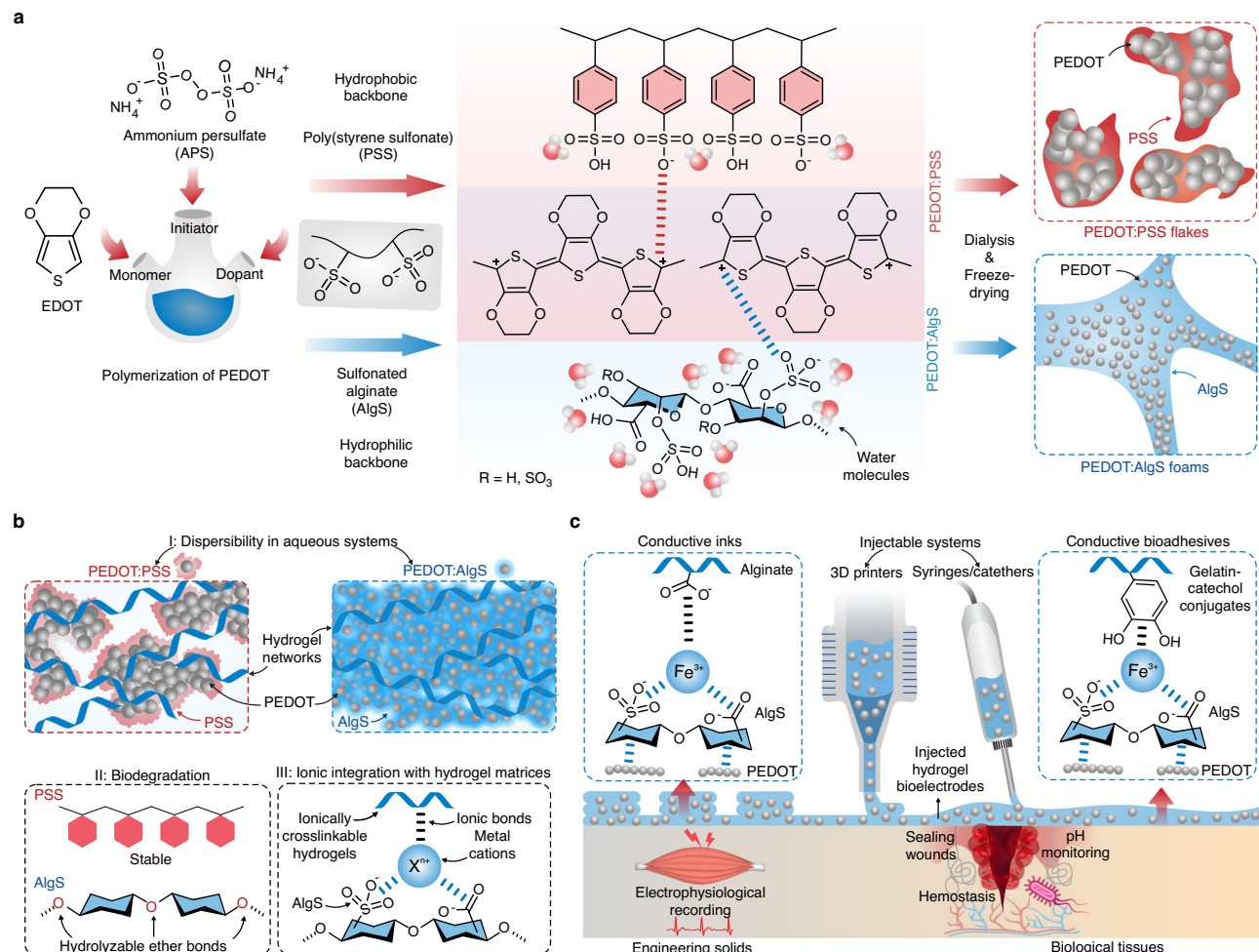

**Fig. 1 | Design and synthesis of water-dispersible poly(3,4-ethylenedioxythiophene) (PEDOT)-based conducting polymers using dopants with hydrophilic backbones. a** Doping of PEDOT via negatively charged macromolecules with hydrophilic and hydrophobic backbones, i.e., sulfonated alginate (AlgS), and poly(styrene sulfonate) (PSS), respectively. **b** Doping of PEDOT with AlgS as compared with PSS leads to enhanced dispersibility in aqueous solutions, improved molecular degradability, and high ionic integrability with ionically cross-linkable hydrogel matrices. **c** The PEDOT:AlgS polymers serve as high-concentration dispersible fillers in hydrogel pre-polymers for the development of injectable and 3D-printable hydrogel bioelectronics for wearable physiological recordings as well as wound closure and monitoring.

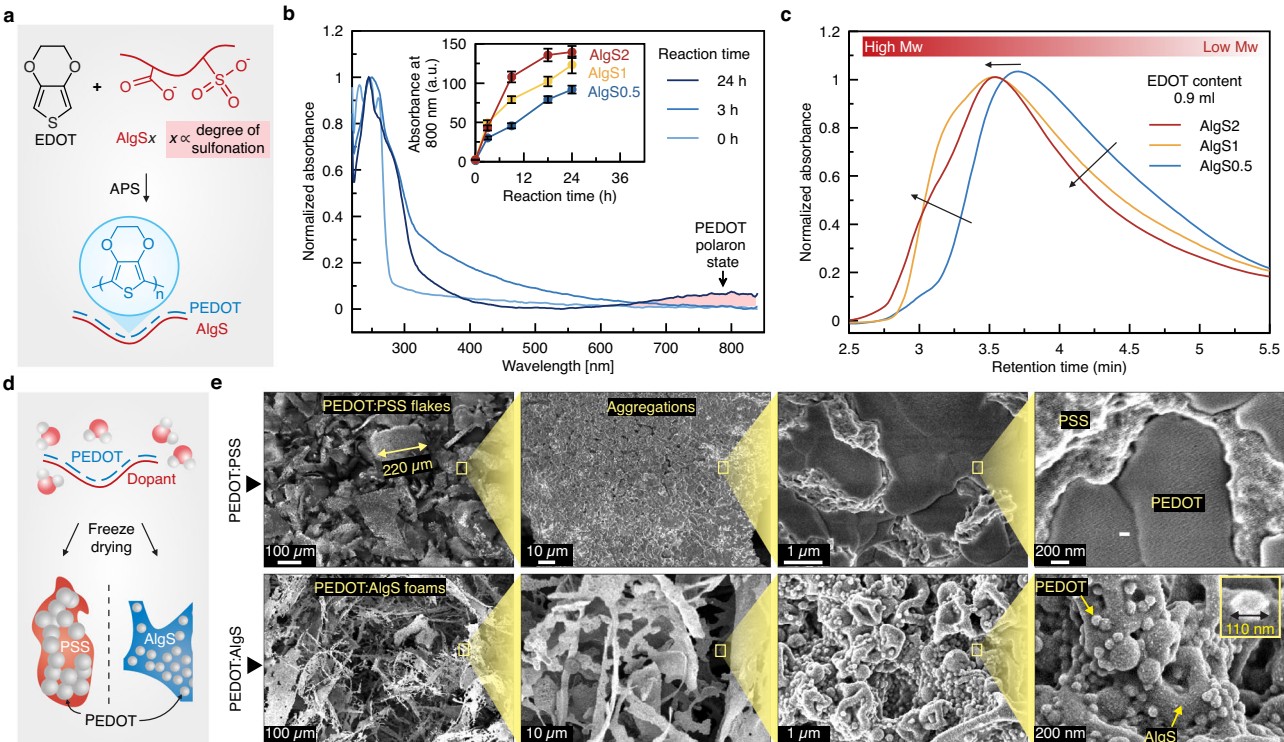

**Fig. 2 | Characterization of freeze-dried sulfonated alginate-doped PEDOT polymers. a** Polymerization and doping of PEDOT. The AlgS samples are labeled as AlgSx where x represents the % w/v concentration of CSA used in sulfonation reaction. **b** UV-vis spectra of PEDOT:AlgS2 at different polymerization times. EDOT 3,4-ethylenedioxythiophene, APS ammonium persulfate. The inset shows the absorbance of PEDOT:AlgS reaction solutions at 800 nm before freeze-drying. Data in the inset is presented as mean ± standard deviation. **c** Effect of alginate

sulfonation on the molecular weight distribution of freeze-dried PEDOT:AlgS2 tested via size-exclusion chromatography (SEC) (PEDOT polymerized for 1 d). **d** Freeze-dried PEDOT forming large aggregates when doped with PSS while doping with AlgS2 results in homogeneously distributed small nanoparticles. **e** Scanning electron microscopy (SEM) images of freeze-dried PEDOT:AlgS2 and PEDOT:PSS. Values represent the mean (n = 3 independent samples).

signals such as physiological outputs and temperature compared to PEDOT:PSS (Fig. 1c). Finally, we showcase the ability of PEDOT:AlgS to introduce pH-sensitive conductivity to gelatin-based bioadhesives for the development of smart sealants capable of wound monitoring applications where the pH sensitivity is enhanced by ~250% compared to conventional PEDOT:PSS. We envision that such PEDOT:AlgS will serve as versatile and biocompatible building blocks, leveraging hydrogels as functional electrodes for soft injectable electronic applications.

## Results

### Design and characterization of water-dispersible PEDOT conductive additives

The synthesis of PEDOT:AlgS consists of a two-step reaction: (1) modification of alginate (W201502, ~200 kDa[36]) with sulfonate groups using chlorosulfonic acid (CSA) to yield negatively charged hydrophilic AlgS dopants (Supplementary Fig. 1a, b); and (2) using obtained AlgS to dope PEDOT during the oxidative polymerization of EDOT, resulting in PEDOT:AlgS conductive polymers (Fig. 1a). The Fourier transform infrared (FTIR) spectra of AlgS (Supplementary Fig. 1c) revealed peaks at 1200 cm⁻¹, indicating the conjugation of sulfonate groups during the first reaction step. The results of sulfonation degree (Supplementary Fig. 1d) suggest approximately 37% conversion of hydroxyl groups in alginate when CSA concentration exceeded 1.5% w/v, which then plateaued. Consequently, a 2% w/v CSA concentration was established as the upper limit for subsequent experiments. Size-exclusion chromatography (SEC) tests (Supplementary Fig. 2a) showed consistent trends, with minor peak shifts to higher retention times as CSA concentration increased, indicating minor chain degradations during sulfonation.

Functionalizing alginate with sulfonate groups substantially improved AlgS's water solubility (Supplementary Fig. 2b). The hydrogel formation capacity of AlgS with multivalent cations was tested at their highest soluble concentrations (Supplementary Fig. 2c, d). Rapid crosslinking with gelation points within seconds after exposure to ionic solutions was recorded. Storage moduli at 5 min indicated that sulfonation prevented ionic crosslinking of alginate through divalent $Ca^{2+}$ cations, but the larger valent $Fe^{3+}$ cations could still form AlgS hydrogels. While sulfonation interrupted alginate ionic associations, possibly due to steric hindrance and conformational changes, comparable mechanical properties could still be achieved by increasing the AlgS content.

Polymerization of PEDOT in PEDOT:AlgS systems, studied by UV-vis (Fig. 2a, b and Supplementary Fig. 3a, b), began with the formation of EDOT dimers and trimers within the first hours, indicated by a sharp peak at 256 nm[37]. The broad absorption band from 400–600 nm corresponded to $\pi \rightarrow \pi^*$ transitions in the neutral state of PEDOT[37]. Near-infrared absorption bands at wavelength ranges of 600–900 nm and 700–1200 nm suggest transitions to the polaronic and bipolaronic states, respectively, due to doping via anionic sulfonate groups of AlgS (Supplementary Fig. 3c)[38]. The results also indicate preservation of the doped state after dialysis and freeze-drying. This trend aligns with the previous reports on PEDOT:PSS polymerization[39]. FTIR spectra of PEDOT:AlgS samples showed peaks at 1358 cm⁻¹ due to C−C and C=C stretching vibrations in quinoidal thiophene rings of PEDOT (Supplementary Fig. 3d)[40], while C−S stretching vibrations produced a strong peak at 984 cm⁻¹. SEC tests exhibited a peak shift to lower retention times with increasing EDOT content and degree of sulfonation (Fig. 2c and Supplementary Fig. 3e), signifying the formation of larger molecular weight PEDOT structures.

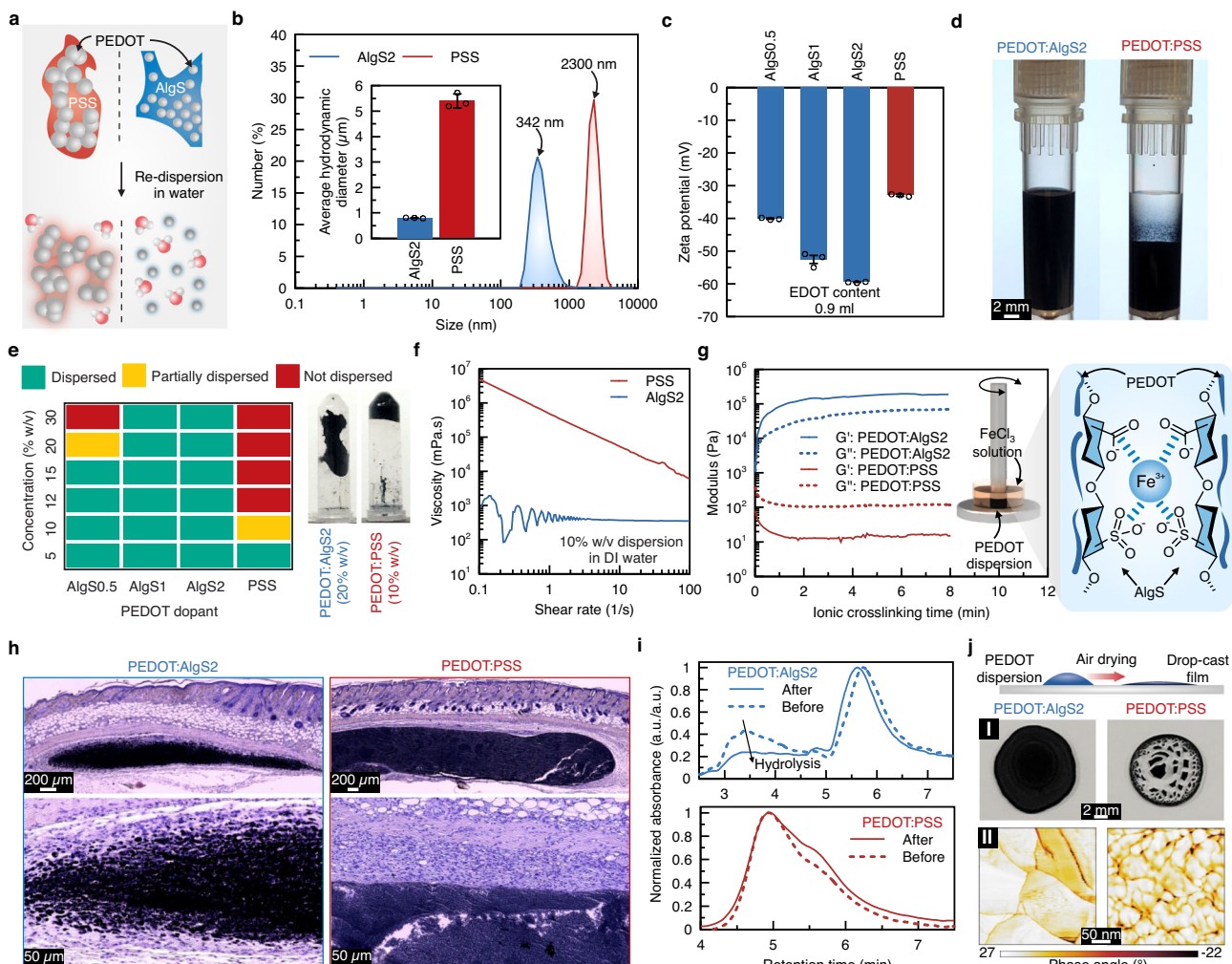

**Fig. 3 | Aqueous re-dispersion properties of PEDOT:AlgS. a** Facilitated re-dispersion of PEDOT in aqueous solutions via hydrophilic AlgS dopants. **b** The number size distribution of PEDOT polymers obtained from dynamic light scattering (DLS) tests. The inset represents the average hydrodynamic sizes of PEDOT re-dispersions in water ($n = 3$ independent samples). **c** Results of zeta potential for aqueous PEDOT:AlgS ($n = 3$ independent samples). **d** Comparison of the colloidal stability of 2% w/v PEDOT:AlgS2 with PEDOT:PSS after 1 week at rest. **e** Dispersibility of freeze-dried PEDOT:AlgS2 in water at various sulfonation degrees and PEDOT:PSS. Tubes represent 10% w/v PEDOT:PSS in water forming self-associated gels and 20% w/v PEDOT:AlgS remaining in liquid phase. **f** Viscosity-shear rate profiles for 10% w/v solutions of PEDOT:AlgS2 and PEDOT:PSS in water. **g** Ionic crosslinking of PEDOT-based polymers with exposure to $Fe^{3+}$ cations in terms of storage (G′) and loss (G″) modulus. **h** Hematoxylin and eosin (H&E) staining of intradermally injected solutions of PEDOT:AlgS2 and PEDOT:PSS (5% w/v) in vivo after 1-week implantation ($n = 4$ independent samples with similar results). **i** Hydrolysis-driven degradability assessment of PEDOT:PSS and PEDOT:AlgS2 tested via SEC. **j** Air dry coating of PEDOT doped with PSS and AlgS on I: glass substrate and their II: atomic force microscopy (AFM) phase plots obtained from PEDOT dispersions (5% w/v) ($n = 3$ independent samples with similar results).

The electrical conductivity of PEDOT:AlgS in the dry state (Supplementary Fig. 4) showed a high correlation with alginate sulfonation degree, implying its doping effect. Given the differences in polymerization kinetics between PSS and AlgS dopants, the polymerization time for PEDOT:AlgS2 was set to 2 days to achieve a dry conductivity comparable to standard PEDOT:PSS controls, which are typically synthesized over one day[38,41]. Solid-state impedance spectroscopy was used to evaluate the effects of sulfonation degree and EDOT content on the AC performance of polymer films (Supplementary Fig. 5)[42]. In the Randles circuit model[43], the ohmic resistance of PEDOT, $R_p$, showed high correlations with sulfonation degree. Overall, the DC and AC electrical characterizations suggest the formation of percolation networks at EDOT contents of greater than 0.5 ml, which is comparable to the formulation of commercially available PEDOT (Clevios PH1000). A higher EDOT content of 0.9 ml was used for subsequent characterizations to demonstrate the capability of AlgS dopants in high-concentration dispersion of

larger PEDOT:dopant ratios in aqueous systems. From a microstructural perspective, scanning electron microscopy (SEM) images (Fig. 2d, e and Supplementary Fig. 6) revealed that freeze-dried PEDOT:PSS aggregated excessively, leading to submillimeter-scale flake particles. In these images, the observed phases (PEDOT, PSS, and AlgS) were identified based on their distinct morphologies, with porous structures corresponding to hydrophilic polymer phases (dopant) and aggregates corresponding to PEDOT. While these results align well with dynamic light scattering (DLS) data (Fig. 3a), further chemical analyses are required to validate the identified phase attributions. In the PEDOT:PSS group, inhomogeneous phase separation of PEDOT and PSS was evident, while freeze-dried PEDOT:AlgS foams formed evenly distributed nanoparticles (~100 nm) within AlgS phase. This nanoscale structure of PEDOT in PEDOT:AlgS facilitates high-concentration re-dispersion of freeze-dried PEDOT in aqueous systems (Fig. 3a), whereas PEDOT loading in PEDOT:PSS is limited due to macroscale steric hindrance between PEDOT:PSS flakes.

## Aqueous re-dispersion of freeze-dried PEDOT polymers

While dialysis and freeze-drying are essential for removing toxic byproducts and re-dispersing PEDOT-based polymers at controlled and high concentrations in aqueous systems, these processes exacerbate PEDOT aggregations[44]. Here, an enhanced dispersibility was achieved by doping with AlgS, which resulted in smaller size and greater hydrophilicity compared to PSS (Fig. 3a). DLS data revealed that PEDOT:AlgS has an order of magnitude smaller size distribution than PEDOT:PSS when synthesized at the same EDOT content of 0.9 ml (Fig. 3b). The hydrodynamic sizes of PEDOT:AlgS increased with higher degrees of sulfonation and EDOT content (Supplementary Fig. 7a) due to the greater extent of polymerization. Zeta potential measurements (Fig. 3c) showed that the net negative charges in PEDOT:AlgS increased with the degree of sulfonation, reaching approximately double those of PEDOT:PSS counterparts. These repulsive forces contribute to much better dispersion stability in water, as illustrated in Fig. 3d. A longer-term investigation over 3 months (Supplementary Fig. 7b) also confirmed the crucial role of sulfonate conjugates in colloidal stability, which is critical for structural uniformity and ink flow in 3D printing applications.

The dispersibility limits of PEDOT:AlgS were improved with sulfonation of alginate (Fig. 3e), exceeding those of PEDOT:PSS by about 4–5×, as the hydrophilic backbone of alginate facilitated interactions with water molecules. This hydrophilicity was reflected in the contact angle results of Supplementary Fig. 7c, where a substantially lower contact angle was obtained in PEDOT:AlgS (26°) compared to PEDOT:PSS (48°). Poor dispersibility of PEDOT:AlgS at low sulfonation degrees (i.e., AlgS0.5) highlighted the critical roles of sulfonate groups in achieving PEDOT aqueous dispersibility. Viscosity-shear rate characteristics (Fig. 3f and Supplementary Fig. 7d, e) showed that solutions of 10% w/v PEDOT:PSS were drastically more viscous than PEDOT:AlgS. Similarly, commercial PEDOT:PSS solutions compared with PEDOT:AlgS synthesized at the same PEDOT to dopant ratio (1:2.5) resulted in ~15× larger viscosity in solutions of similar concentrations (1.3% w/v) as shown in Supplementary Fig. 7f. This result not only indicates better dispersibility of PEDOT:AlgS, but also suggests that AlgS dopants allow for more PEDOT:dopant ratios compared to PSS. The decreasing trends of viscosity-shear rate curves showed shear-thinning properties of PEDOT solutions, typical of electrostatic hydrogels, which are indicative of their injectability[45].

Evaluation of ionic crosslinkability using alginate-based dopants suggested that PEDOT:AlgS is ionically responsive to $Fe^{3+}$, a response not observed in PEDOT:PSS (Fig. 3g and Supplementary Fig. 8).

## Biocompatibility and biodegradability of PEDOT solutions

In vivo injection of 5% w/v PEDOT:PSS solutions resulted in the formation of a fibrous capsule around PEDOT:PSS within a week after implantation (Fig. 3h), whereas the immune cells infiltrated PEDOT:AlgS samples, attempting to digest the polymers. We attribute the fibrotic capsules around PEDOT:PSS to its higher dispersion viscosity preventing cells from infiltrations. Extended implantation over 11 weeks showed progressive degradation in PEDOT:AlgS with cell infiltration, while PEDOT:PSS remained stable within the fibrotic capsule (Supplementary Fig. 9a, b). No meaningful differences in follicles or accumulation of fatty tissue were seen between the two groups. Immunostaining results suggest the limited presence of macrophages (F4/80+), neutrophils (Ly6G+), and T cells (CD3+) involved among the infiltrated cells in PEDOT:AlgS (Supplementary Fig. 9c). Although, the number of immune cells constituted a substantially lower ratio of the present cells in PEDOT:AlgS compared to the PEDOT:PSS, which implies a stronger immune response in PEDOT:PSS (Supplementary Fig. 9d).

To understand the degradation mechanisms, hydrolysis-driven molecular weight changes of the polymers were tested in vitro (Fig. 3i). The decay seen in the peak of PEDOT:AlgS, which is absent in PEDOT:PSS suggests that, unlike PEDOT:PSS, PEDOT:AlgS is hydrolytically degradable. The enhanced molecular degradability of PEDOT:AlgS is attributed to the hydrolyzable glycosidic bonds on the backbone of AlgS, which are absent in the PSS structure (see Supplementary Fig. 10). While PEDOT is generally stable, the byproducts of alginate are primarily alginate backbone broken into oligo- and monosaccharides involving sulfonated mannuronic acid (M) and guluronic acid (G) residues. The PEDOT phase is expected to remain intact due to the stable bonds, however, its smaller size distribution can promote their renal clearance in vivo. While sulfonation increases the hydrolyzability of alginate, as confirmed by Supplementary Fig. 10b (aligning with previous studies[46]), the enhanced solubility of alginate due to sulfonation can further facilitate its removal from the body. Metabolic pathways of these degradation byproducts are expected to be primarily through renal excretion. We note that although human enzymes do not metabolize sulfonated oligosaccharides, certain bacteria in the gut produce alginate lyases[47], which may further contribute to alginate degradation through enzymatic cleavage of the glycosidic bonds. This biodegradability makes PEDOT:AlgS suitable for transient implantable bioelectronics.

## Conductive polymer coatings

Crack formation during drying has been a major challenge in drop-cast coatings of PEDOT:PSS[48]. Coating on solid surfaces showed an increased coverage ratio with polymer concentration for both PEDOT:PSS and PEDOT:AlgS up to 2.5% w/v (Supplementary Fig. 11a, b). Further increases in PEDOT:PSS concentration beyond 2.5% w/v however, led to prominent cracking and islet formation (Fig. 3j), while PEDOT:AlgS achieved nearly complete surface coverage with minimal defects. The surface morphology and phase distribution of coatings (formed by the 2.5% w/v solutions) (Supplementary Fig. 11c–e) suggested larger grain sizes and a more homogeneous distribution of PEDOT (light region) within the dopant (dark region) in PEDOT:AlgS compared to PEDOT:PSS. Despite larger grain sizes typically being attributed to better conductivity[49] (due to the fewer boundaries and energy barriers), the greater phase separation and thereby interconnectivity of the PEDOT phase in PEDOT:PSS resulted in a comparable conductivity with PEDOT:AlgS (Supplementary Fig. 4a).

## Hydrogel-based 3D-printed soft bioelectrodes

Given its excellent water dispersibility and solution stability, PEDOT:AlgS offers great potential for injectable applications such as 3D-printed bioelectronics (Fig. 4a). The highest dispersible amount of PEDOT:PSS in alginate solutions for inks to remain injectable was approximately 4% w/v, translating to an ~8× possible improvement in conductivity of alginate solutions (Fig. 4b, c). This figure reached ~160× when PEDOT:AlgS was used (~20× greater than PEDOT:PSS) due to its much higher dispersion limit of up to ~20% w/v. The relative conductivity changes with incorporating PEDOT into various hydrogels reported previously (Supplementary Fig. 12) show that relative improvement in conductivity PEDOT:AlgS is remarkably higher, by 1–2 orders of magnitude, among PEDOT-based injectable hydrogels. Here, we emphasize relative conductivity changes with respect to hydrogel matrix to highlight the roles of PEDOT in ohmic conductivity, excluding the effects of ionic conduction and secondary dopants. Given the electrically insulating nature of existing hydrogels (e.g., alginate with a conductivity of $\sim 7.1 \times 10^{-4}$ S $m^{-1}$), the conductivity of PEDOT:AlgS-incorporated hydrogels ($\sim 7.5 \times 10^{-2}$ S $m^{-1}$) is lower than reports on pure PEDOT hydrogels (on the order of $10^{-3}$–$10^{-5}$ S $m^{-1}$[22,25]). However, it is important to note that processing pure PEDOT hydrogels typically requires drying steps and the use of organic solvents or

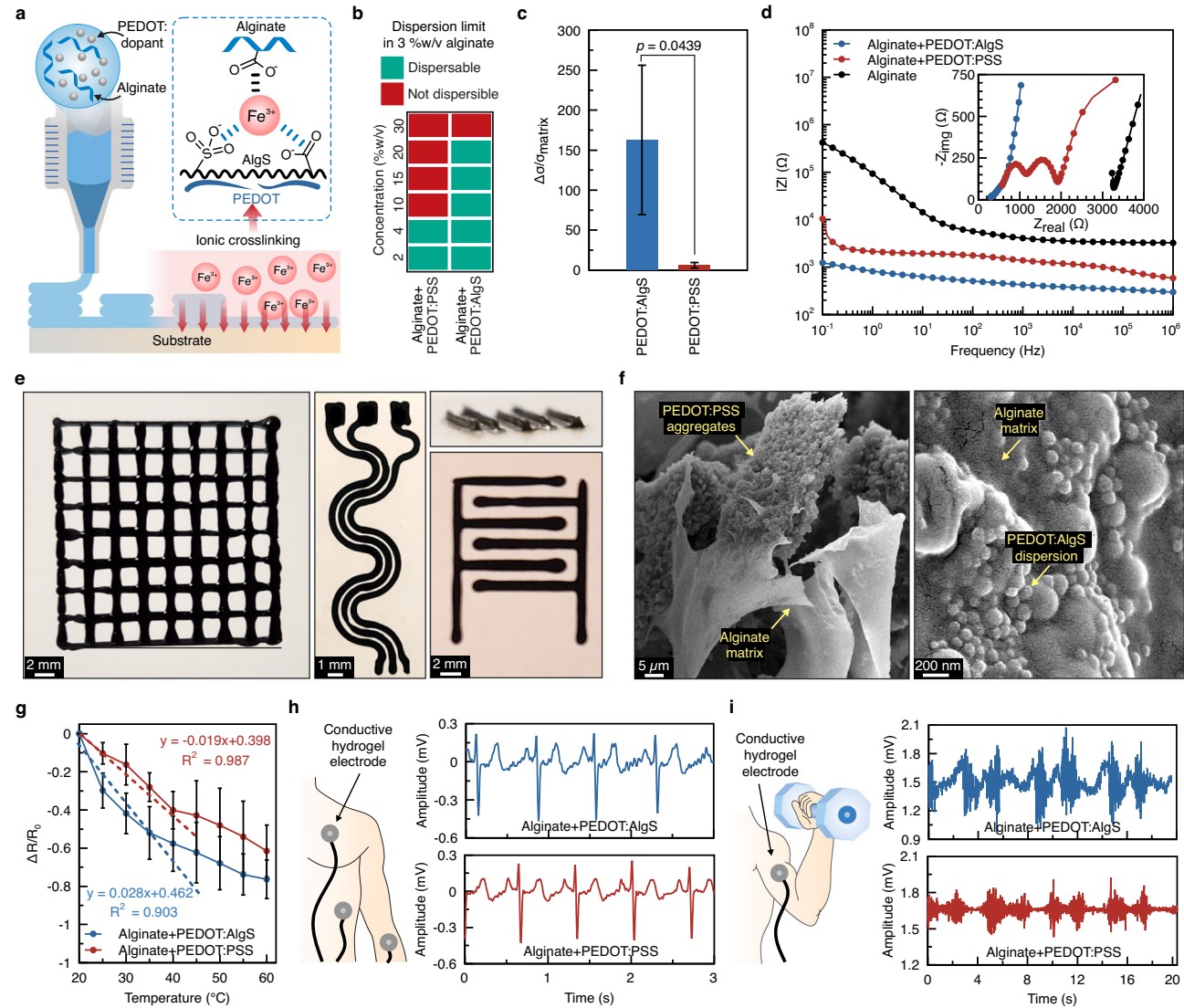

**Fig. 4 | 3D-printable conductive PEDOT:AlgS inks for wearable sensing.**
**a** Formulation and crosslinking scheme of conductive inks comprised of alginate with PEDOT fillers doped with AlgS and PSS. **b** Dispersibility limit of PEDOT:AlgS and PEDOT:PSS in 3% w/v alginate solutions in water. **c** The relative increase in conductivity of alginate ($7.1 \times 10^{-4}$ S m$^{-1}$) as a result of introducing conductive PEDOT fillers at their dispersibility limit before crosslinking with $Fe^{3+}$. The $p$-value is determined from a two-tailed Student's $t$-test with unequal variances ($n = 3$ independent samples). **d** Effect of PEDOT dopants on the impedance spectroscopy characteristics of alginate inks. The inset shows Nyquist plots corresponding to the ink formulations. **e** Illustration of the multilayer patterns of PEDOT:AlgS in alginate after crosslinking in 25 mM FeCl$_3$ solutions. **f** SEM images of the freeze-dried hydrogels based on the mixtures of PEDOT:PSS and PEDOT:AlgS (4 and 20% w/v, respectively) in alginate ($n = 3$ independent samples with similar results). **g** Temperature sensitivity of PEDOT-incorporated alginate hydrogels ($n = 3$ independent samples). **h, i** Results of electrocardiogram (ECG) and electromyography (EMG) recordings using conductive hydrogels as electrode interfaces, respectively. Values in (**c**) and (**g**) represent the mean and the standard deviation ($n = 3$ independent samples).

cytotoxic ions, which restricts their applicability in scenarios where direct injectability is required.

A more in-depth analysis of the charge-carrying processes was performed via electrochemical impedance spectroscopy (EIS) (Fig. 4d and Supplementary Fig. 13). Nyquist plots showed that alginate-PEDOT:AlgS intersects with $Z_{real}$ at lower impedances compared with alginate-PEDOT:PSS, reiterating greater electrical conductance. Conductivity through PEDOT-based polymers consisted of direct charge transfer through percolation resistance $R_p$, which was ~3× larger in alginate-PEDOT:PSS compared to alginate-PEDOT:AlgS.

In terms of printing fidelity (Fig. 4e), solutions of 4% w/v PEDOT:PSS in alginate experienced multiple clogging events due to aggregation and precipitation (Supplementary Fig. 14a), whereas PEDOT:AlgS solutions (at 20% w/v concentration) were continuously deposited with no visible defects. Crosslinking of 3D-printed

constructs was performed in FeCl$_3$ solutions at its cytocompatible concentration limit of 25 mM (Supplementary Fig. 15). The distribution and morphology of PEDOT:PSS and PEDOT:AlgS after integration with alginate (Fig. 4f and Supplementary Fig. 14b, c) show aggregation of PEDOT:PSS within the alginate matrix, whereas PEDOT:AlgS formed a highly homogeneous and uniform dispersion within the alginate network, explaining the observed conductivity characteristics.

The potential applications of the conductive alginate inks were studied as temperature-sensing elements in medical devices (Fig. 4g). The temperature sensitivity of the alginate with PEDOT:AlgS within the physiologically relevant ranges (20–40 °C) was found to be ~75% greater than those of PEDOT:PSS-incorporated alginate. Additionally, as a proof-of-concept, we explored the capability of alginate-based conductive electrodes in electrophysiological

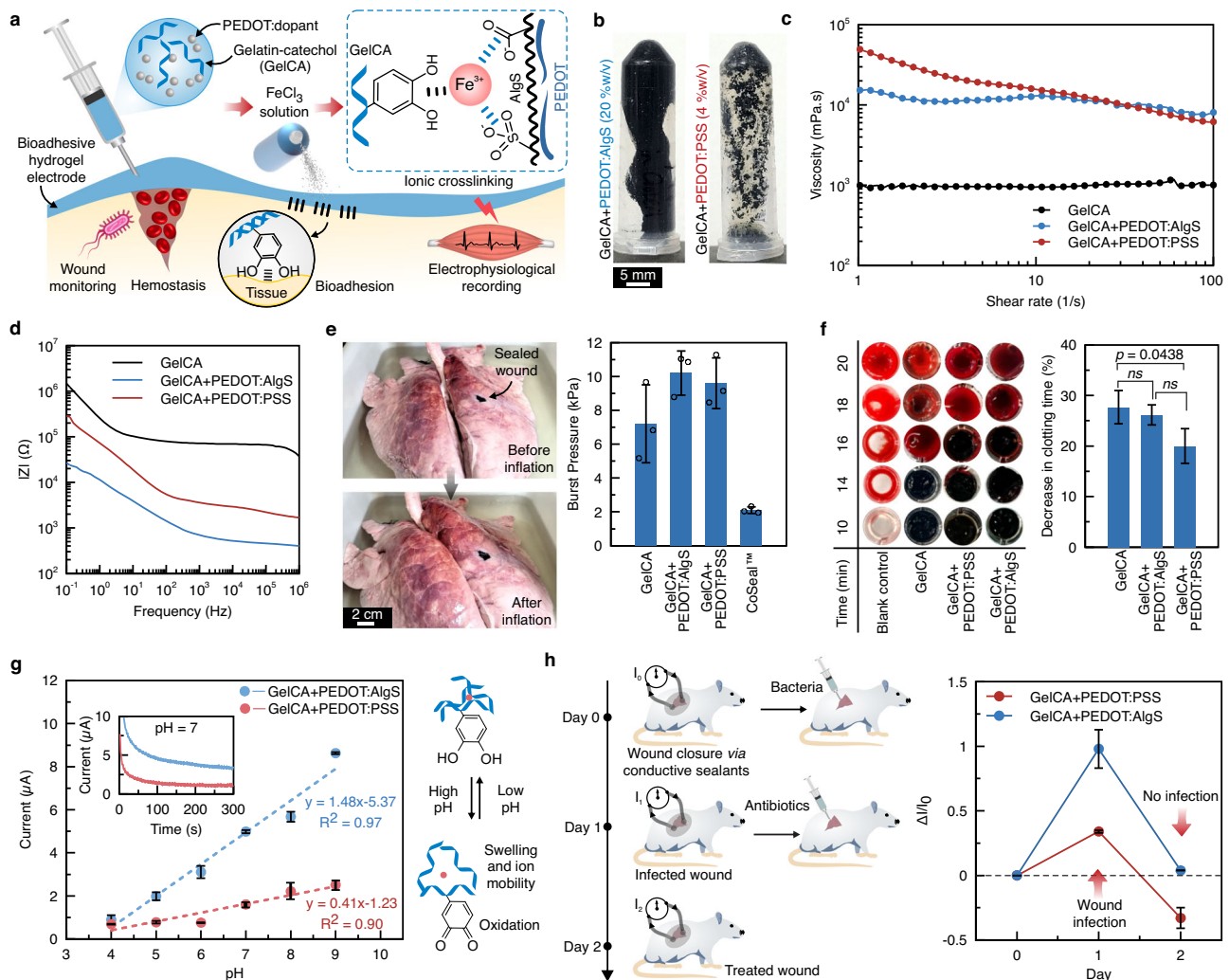

**Fig. 5 | Injectable conductive bioadhesives for implantable bioelectronics applications. a** Formulation of bioadhesives involving gelatin-catechol (GelCA), synthesized via coupling caffeic acid to gelatin, as hydrogel bioadhesive matrices. PEDOT doped with PSS and AlgS are incorporated separately within GelCA at their dispersibility limits (4 and 20% w/v, respectively) and crosslinked ionically by $Fe^{3+}$ for pH sensing in wound monitoring applications. **b** Hydrogel pre-polymer composites of GelCA (12% w/v) with PEDOT:PSS (4% w/v) and PEDOT:AlgS (20% w/v) after shaking on a vortex for 1 min. **c** Viscosity-shear rate of injectable bioadhesives. **d** Impedance spectroscopy of injectable bioadhesive pre-polymers. **e** Ex vivo porcine lung burst pressure adhesion testing of hydrogels (n = 3 independent samples). **f** Clotting time assays in terms of relative decrease in coagulation time for the assessment of hemostatic activity after hydrogel crosslinking. Clotting time for blank controls was 21.7 ± 0.6 min. Statistical analysis was performed via one-way ANOVA (n = 3 independent samples). **g** In vitro pH sensitivity of conductive bioadhesives obtained by chronoamperometric testing of hydrogels in various pH levels. The inset shows current variations with time at pH 7 (n = 3 independent samples). **h** In vivo monitoring of wound infection using conductive bioadhesive hydrogels. The data in (**e**–**h**) represents the mean and the standard deviation (n = 3 independent samples).

recording. Electrocardiography (ECG) measurements via electrodes attached to the volunteer's wrist (Fig. 4h) resulted in time delays between two contiguous S waves of 0.71 and 0.69 s for electrodes containing PEDOT:AlgS and PEDOT:PSS, respectively. These figures correspond to heart rates of 84 and 87 beats min⁻¹, which fall within the healthy regime of 60–100 beats min⁻¹. While the potential amplitudes for S waves were found to be similar, these amplitudes for T waves were found to be approximately 30% greater than those observed for PEDOT:PSS. Similarly, electromyography (EMG) signals resulted in ~43% greater signal amplitudes when lifting 13 lb weights using PEDOT:AlgS-based electrodes compared to those of PEDOT:PSS (Fig. 4i), suggesting the potential of the proposed bioelectrodes for biomonitoring interfaces. This potential was further confirmed by the evaluation of immunoactivity for implantable applications in vitro, where results suggested no inflammatory response associated with PEDOT doped with either PSS or AlgS (Supplementary Fig. 16).

## Injectable smart bioadhesives for wound monitoring

Injectable bioadhesive hydrogels have enabled robust sealing of bleeding wounds[50,51]. However, septic control post-wound closure remains a major challenge as it requires continuous real-time monitoring of patients[52,53]. Incorporating pH-sensing elements such as PEDOT-based polymers[54] to bioadhesives is a promising approach to ensure early detection of potential infections and prompt medical intervention. PEDOT polymers were integrated with a biodegradable and ionically cross-linkable bioadhesive platform (catechol-modified gelatin-caffeic acid conjugates, GelCA)[55] as shown in Fig. 5a. The dispersion limits of PEDOT:AlgS and PEDOT:PSS in GelCA were similar to those reported above for alginate matrices—20% w/v and 4% w/v, respectively—highlighting the substantially better dispersibility of PEDOT:AlgS in aqueous hydrogel systems. Incorporating PEDOT:PSS in GelCA, although at a lower concentration, required much more rigorous mixing than PEDOT:AlgS to attain a homogeneous solution (Fig. 5b). We observed an increase in viscosity (Fig. 5c) and gel-sol

transition temperature (Supplementary Fig. 17) of GelCA with the addition of 20% w/v PEDOT:AlgS, and these increases were comparable to 4% w/v PEDOT:PSS in GelCA. These effects are primarily due to the intensified dynamic interactions, such as thermosensitive hydrogen bonding in GelCA networks[56].

In terms of tensile mechanical properties (Supplementary Fig. 18a), the elastic modulus of GelCA increased more substantially with the addition of PEDOT:AlgS compared to PEDOT:PSS (Supplementary Fig. 18b) at their dispersibility limits, due to denser dynamic interactions introduced at high concentrations of PEDOT:AlgS. The increase in tensile strength with PEDOT:AlgS was also more prominent compared to PEDOT:PSS (Supplementary Fig. 18c), though no substantial difference was seen across all conditions in terms of stretchability (Supplementary Fig. 18d).

Supplementary Fig. 19a represents the conductivity of the pre-gel solutions. While PEDOT:PSS resulted in a maximum of ~1.5× improvement at its dispersion limit, PEDOT:AlgS showed a considerably larger improvement (~7×) due to its superior dispersibility. Crosslinking of PEDOT:AlgS hydrogels using $FeCl_3$ further increased this figure to ~9.5×, which could be elevated up to over 35× depending on the crosslinking time. The EIS results for PEDOT-incorporated GelCA solutions, along with the equivalent circuit constants (Fig. 5d and Supplementary Fig. 19b–d), corroborate the greater capacity of PEDOT-AlgS additives to impart conductivity to GelCA hydrogels.

The GelCA matrix concentrations in hydrogels were engineered to yield stable wet adhesion to collagen sheet tissue models (Supplementary Fig. 20a). Raising the GelCA content to 12% w/v allowed the formation of stable crosslinked hydrogels that remained integral and adhered to collagen substrates. Swelling tests suggested substantial swelling in the GelCA and GelCA+PEDOT:PSS groups, while no swelling was observed in the GelCA+PEDOT:AlgS groups possibly due to the larger content of hydrophobic PEDOT moieties and denser ionic crosslinking enabled by AlgS (Supplementary Fig. 20b). As shown in Fig. 5e, adding either PEDOT:PSS or PEDOT:AlgS to GelCA led to substantial improvements to ex vivo adhesion performance (porcine lungs burst pressures), primarily due to enhanced hydrogel cohesion. Such conductive bioadhesives showed up to 5-fold greater burst pressure than commercial sealants. In addition to physical sealing, the hemostatic efficacy of bioadhesives is crucial for bleeding control and wound healing[57]. Clotting time assays (Fig. 5f and Supplementary Fig. 21) suggested that all hydrogels exhibited hemostatic potency, primarily attributed to the $Fe^{3+}$ ions used for crosslinking and the high density of electrostatic interactions driven by the positive and negative charges of PEDOT additives[58].

The pH sensing function of the hydrogels was tested in physiologically relevant pH ranges[59] (Supplementary Fig. 22a, b). Chronoamperometric studies showed that current increased with pH where the current changes were substantially greater for GelCA-PEDOT:AlgS compared to GelCA-PEDOT:PSS, translating into ~250% improvement in overall sensitivity (Fig. 5g). This enhancement is due to PEDOT:AlgS facilitating the incorporation of larger amounts of pH-responsive PEDOT moieties into the hydrogel. The open circuit potential (OCP) data supported the chronoamperometric measurements (Supplementary Fig. 22c). The pH sensing in conductive hydrogels is generally attributed to the synergistic effects of mediated electron and ion mobility, facilitated by hydrogel swelling, as well as catechol-to-quinone transitions induced by oxidation reactions[54,60,61]. However, we note that swelling effect was found to be negligible in GelCA-PEDOT:AlgS, whereas it was more prominent under other conditions (Supplementary Fig. 20b). Stability of conductivity over 1 week in various pH levels suggested the negligible effects of non-reversible reactions such as catechol oxidation (Supplementary Fig. 22d). In vivo application of conductive bioadhesives in monitoring wound infection demonstrated that PEDOT:AlgS enables ~3× larger relative current change in response to infection compared to PEDOT:PSS-based bioadhesives (Fig. 5h). This resistance change was fully recoverable with antibiotic treatment in PEDOT:AlgS, showing the robust capacity of the PEDOT:AlgS additives for monitoring wound conditions. The current response to wound infection was also tested 3 days after introducing skin wounds where similar trends were observed, indicating sensing sustainability in wound healing timescales (Supplementary Fig. 22e).

Live/Dead imaging results (Supplementary Fig. 23a, b) indicated that both PEDOT:AlgS (at 5× higher concentration) and PEDOT:PSS did not induce any evident cytotoxicity in GelCA hydrogels. Accordingly, the addition of PEDOT-based additives did not affect cell proliferation (Supplementary Fig. 23c), supporting the safety of the synthesized products for potential implantable applications. Likewise, PEDOT additives did not show substantial antibacterial effects nor affect wound healing properties in vivo (Supplementary Fig. 24).

## Discussion

PEDOT:PSS is a popular conductive filler for enhancing conductivity in hydrogels for medical devices and applications, including injectable soft bioelectronics for minimally invasive therapy. Doping conditions play a pivotal role in the characteristics of these polymers, such as electrical properties and aqueous dispersion. Doping PEDOT via hydrophilic alternatives is an effective strategy to enhance the dispersion limit of PEDOT in aqueous systems, allowing dispersions of more than 20% w/v in hydrogels, which is over 5× greater than those of PEDOT:PSS. This approach enables increasing conductivity thresholds up to 20× beyond those achievable by PEDOT:PSS fillers at their dispersibility limit due to the formation of more compact percolation networks. Additionally, the long-term dispersion stability of PEDOT was markedly improved, rendering PEDOT:AlgS stable additives for long-term ink storage and applications in 3D printing. Unlike PEDOT:PSS, ionic associations are possible with PEDOT:AlgS (due to using alginate derivatives as dopants) in response to $Fe^{3+}$ cations, suggesting opportunities for stronger integration with other ionically crosslinking hydrogel systems such as alginate and catecholic bioadhesive hydrogels. These properties, combined with excellent in vivo response due to mitigation of fibrotic capsule formation, make PEDOT:AlgS an ideal candidate for implantable, biodegradable, and injectable bioelectronics.

In combination with hydrogel systems, introducing electrical conductivity to gelatin-based bioadhesives via PEDOT:AlgS enables smart sealants capable of pH monitoring for sensing wound conditions such as infection. Since PEDOT:AlgS enables a greater dispersion limit, the pH sensitivity in their corresponding electrodes is enhanced substantially by ~250%.

Overall, doping conducting polymers with hydrophilic moieties such as AlgS holds promising potential for applications in injectable bioelectronics. We envision that various natural biomolecules can be modified for doping polymeric semiconductors to design bioactive electrodes that enable more efficient interfacing with tissues and engineering biomaterials such as soft hydrogels. Integrability of these additives with other crosslinking systems, such as free-radical photopolymerization, further expands their utility in the development of minimally invasive theranostics. Besides, despite advances in improving conductivity, hydrogel-based electrodes still exhibit substantially lower conductivity compared to metallic electrodes, highlighting the need for further innovations in this space. Continued efforts in this field will open new avenues for sophisticated injectable bioelectronic systems capable of long-term in vivo monitoring and therapeutic interventions for more applications in real-time health monitoring and neural interfacing. Lastly, long-term in vivo studies of immune response and degradation, along with demonstrations of PEDOT-based hydrogels with active wound healing and antimicrobial properties, are critical to expand their application for wound monitoring.

## Methods

### Materials and reagents

Type A gelatin derived from porcine skin, gelatin from cold water fish skin, sodium alginate (cat. no. W201502), calcium chloride ($CaCl_2$), iron(III) chloride ($FeCl_3$), caffeic acid (CA), sodium chloride (NaCl), ammonium persulfate (APS), poly(sodium 4-styrenesulfonate) (PSS) average molecular weight 70,000 Da, formamide, dimethyl sulfoxide (DMSO), chlorosulfonic acid (CSA), hydrochloric acid (HCl), and sodium periodate ($NaIO_4$) were acquired from Sigma-Aldrich. 1-Ethyl-3-(3-dimethylaminopropyl)carbodiimide (EDC), $N$-hydroxysuccinimide (NHS), and 3,4-ethylenedioxythiophene (EDOT) were purchased from TCI Chemicals, USA. Porcine lung tissues were provided by Sierra for Medical Science, USA. Dulbecco's phosphate-buffered saline (PBS) and buffer solutions at pH 4, 7, and 9 were supplied by ThermoFisher Scientific, USA. Ethanol ($C_2H_5OH$) was purchased from KOPTEC (King of Prussia, PA, USA), potassium phosphate monobasic ($KH_2PO_4$), and potassium phosphate dibasic ($K_2HPO_4$) were purchased from Sigma-Aldrich, USA. The commercial PEDOT:PSS (1.3 wt.% dispersion in $H_2O$) was supplied by Sigma-Aldrich, USA.

### Synthesis of PEDOT:AlgS

Alginate was modified to obtain AlgS dopants. First, 1000 mg alginate was added to 40 ml formamide and placed in an ice bath while stirring at 250 rpm. Different batches of AlgS$x$ were synthesized by adding various amounts of CSA (i.e., $x$ = 0.5, 1, and 2 ml) to the reaction solutions, and the reactions proceeded for 4 h at 60 °C under stirring at 250 rpm. Then, AlgS was precipitated by adding an equal volume of acetone to the solution. The solution was vortexed and centrifuged at 3000×$g$ for 5 min to form a pellet. The supernatant was discarded, and the AlgS pellets were dissolved in deionized (DI) water at ~2% w/v. The AlgS solution was dialyzed for 24 h against DI water, where the dialysis media was refreshed 3 times and subsequently freeze-dried.

To synthesize PEDOT:AlgS, 1000 mg AlgS was dissolved in 50 ml DI water while stirring at 250 rpm. Then, different amounts of EDOT (i.e., 0.1, 0.5, and 0.9 ml) and APS (i.e., 240, 1200, and 2160 mg corresponding to each EDOT concentration) were added to the mixture. Where not specified, PEDOT:AlgS is synthesized using AlgS2 and the highest EDOT concentration. The polymerization reaction proceeded at room temperature at 250 rpm for 48 h unless noted otherwise.

Then, the reaction solutions were dialyzed for 3 days against 100 mM NaCl (the dialysis media refreshed 3 times a day), followed by freeze-drying. All products were dialyzed and freeze-dried (1) to remove ions/byproducts for biocompatible interfacing with tissues and (2) to allow re-dispersion at controlled concentrations.

### Synthesis of PEDOT:PSS

The synthesis of PEDOT:PSS controls was performed following the same protocol as PEDOT:AlgS, where doping of PEDOT was conducted by equivalent weight ratios of PSS to EDOT monomers (i.e., the amount of EDOT monomers and APS initiators were 0.9 ml and 2160 mg, respectively). Polymerization of PEDOT in PEDOT:PSS was conducted for 24 h (half of the EDOT polymerization time in PEDOT:AlgS) to yield similar electrical conductivity between the two conditions.

### Synthesis of bioadhesive gelatin-catechol conjugates

For modification of gelatin with bioadhesive catechol motifs (gelatin-catechol, GelCA), 212 mg CA (4.25% w/v) was dissolved in a 5 ml solution of 50 %v/v DMSO in DI water. Subsequently, 7.5 mg of $NaIO_4$ (0.15% w/v) was added to the CA solution, and the solution was stirred for 1 h to produce oxidized CA. Following this step, a solution consisting of 20 ml of 50 %v/v DMSO in water along with 191 mg of EDC (0.96% w/v) and 115 mg of NHS (0.58% w/v) was introduced into the mixture and activation reaction was allowed to proceed for 2 h.

Simultaneously, 1 g of gelatin is dissolved in a 75 ml solution of 50% v/v DMSO in water (1.33% w/v) at 50 °C for 1 h. The gelatin solutions were mixed with NHS-activated oxidized CA solution, and the conjugation reaction was left to proceed for 24 h at room temperature. Subsequently, the resulting solutions are dialyzed in DI water for 3 days, with the dialysis solution being refreshed three times daily. Finally, the samples were subjected to freeze-drying for 3 days.

### Hydrogel formation

Hydrogel mixtures containing PEDOT-based polymers were cross-linked through ionic crosslinking. Alginate inks were prepared by mixing PEDOT:PSS (4% w/v) and PEDOT:AlgS (20% w/v) in 3% w/v sodium alginate. The inks were exposed to 25 mM $FeCl_3$ solution for 5 min and washed with PBS. Bioadhesive hydrogels were prepared similarly by mixing equivalent amounts of PEDOT:PSS and PEDOT:AlgS in 12% w/v GelCA solutions in water at 80 °C followed by shaking until complete dissolution. The pre-polymers were then injected at room temperature, leading to their physical gelation and then, ionic cross-linking was conducted via $FeCl_3$ diffusion (soaking in 25 mM $FeCl_3$ solution for 5 min).

### Solubility measurements

The solubility/dispersibility in water was assessed by mixing dry components in DI water at various concentrations, followed by multiple stages of shaking (by vortex) and heating at 80 °C. These samples were then fixed at an angle and held in position for 10 s where the flowability of the solutions under gravity was correlated to stability. The samples were considered dispersible if they flew under gravity and non-dispersible if did not.

### Swelling tests

Water absorption of hydrogels was assessed via swelling experiments. Hydrogels were crosslinked and their initial wet weight was recorded. They were then soaked in DI water at room temperature and their weight was registered at different time points. The swelling ratio was calculated as the ratio of weight change to initial weight.

### UV-vis spectroscopy

The UV-vis measurements were performed using a NanoDrop OneC microvolume spectrophotometer (ThermoFisher Scientific, USA) on 2 μl volumes taken from the sample solutions. For UV-vis analysis of PEDOT:AlgS after freeze-drying, the polymers were dispersed at 1% w/v in DI water. For the analysis of polymerization kinetics, samples were taken directly from the reaction solution at different time points. UV-vis measurements in the near-infrared region were performed using an Agilent 8453 UV-vis Spectrometer.

### Fourier transform infrared spectroscopy

The FTIR spectroscopy measurements were conducted via attenuated total reflection (ATR)-FTIR on freeze-dried AlgS and PEDOT:AlgS samples using a Bruker Alpha II Platinum ATR spectrometer. Baseline correction was performed on the data using SpectraGryph software.

### Size-exclusion chromatography

The molecular weight distribution of the samples was assessed using SEC tests. The samples were dissolved in DI water at 0.5 mg/ml and filtered through 0.8 μm filter papers. For accelerated hydrolytic degradability tests, the products were treated in 20 mM NaOH at 5% w/v concentration and maintained at 37 °C for 3 weeks. Before the SEC tests, the samples were diluted to 0.5 mg ml$^{-1}$ using DI water. Samples were loaded into an Agilent 1200 high-performance liquid chromatography (HPLC) equipped with a UV detector. A Yarra 3 μm SEC-3000 column (Phenomenex, Torrance, CA, USA) at 30 min running time was used with the mobile phase being PBS. The absorbance data measured at 210 nm wavelength was reported over retention times.

## Dynamic light scattering

A Nano ZS (Malvern Panalytical Ltd, UK) zeta sizer was used for the DLS tests for the characterization of zeta potential and size distributions of the dispersions. The test solutions were prepared at 0.5 mg/ml, followed by loading into the DTS1070 disposable folded capillary cuvettes. The refractive indices for the dispersant and polymers, as well as the polymer absorption constant, were defined as 1.33, 1.51, and 0.01, respectively. The data were recorded at room temperature.

## Contact angle measurements

Contact angle measurements were performed to assess the wettability of the surfaces using a goniometer RemaHart. A volume of 100 μl PEDOT solution in water (2.5 % w/v) was dried on glass slides on hotplates at 60 °C. A droplet of deionized water (~5 μl) was carefully placed on the surface of coated layer using a microsyringe. Images of the droplet were captured upon contacting surface using a high-resolution camera, and the contact angle was determined by analyzing the droplet profile using ImageJ software.

## Scanning electron microscopy

To analyze microstructural features, SEM images were captured. Prior to imaging, a thin layer of gold (20 nm) was sputtered onto the surface of each sample. The images were recorded at 2 kV acceleration voltage using a Zeiss Sigma 300 SEM microscope.

## Atomic force microscopy

To analyze the phase images and surface roughness, atomic force microscopy (AFM) was conducted using a Dimension® Icon® atomic force microscope with ScanAsyst™. The PEDOT-based coatings were prepared by drop casting 75 μl of 5% w/v solutions on a 60 °C hotplate. Imaging was performed at 300 nm scan size and 1.5 Hz scan rate.

## Degree of sulfonation

The sulfonation degree of AlgS was quantified using barium sulfate nephelometry[62]. First, 15 mg AlgS underwent hydrolysis in 5 ml of 0.1 M HCl at 120 °C for 7 h to release the attached sulfonate groups. Subsequently, the solution volume was adjusted to 25 ml by adding DI water. Then, 625 μl of this solution was mixed with 312.5 μl of a gelatin-barium chloride solution consisting of 5% w/v BaCl$_2$ and 5% w/v gelatin from cold water fish skin in DI water. The mixture was added to 175 μl of 8% w/v trichloroacetic acid in DI water and thoroughly mixed before allowing the reaction to proceed for 20 min. The absorbance of the reaction solutions was measured at 350 nm wavelength using a microplate reader (BioTek UV-vis Synergy 2, VT, USA). The degree of sulfonation was measured based on the calibration curves.

To attain standard curves (absorbance-SO$_3$ concentration), stock sodium sulfate (NaSO$_4$) solutions of various concentrations, i.e., 2.500–0.005 mg/ml, were prepared in DI water. Then, 500 μl of this solution was mixed with 250 μl gelatin-barium chloride and 140 μl trichloroacetic acid solutions prepared previously. The absorbance measurement was performed accordingly, and the degree of sulfonation was measured using the Eq. (1):

$$DS = \frac{88.06 \times [S_{AlgS}]}{0.39 - 80.07 \times [S_{AlgS}]} \qquad (1)$$

where DS is the degree of sulfonation, and $S_{AlgS}$ is the corresponding concentration of sulfonate groups for AlgS samples (in mol l$^{-1}$) as obtained from the standard curves.

## Rheological tests

Rheological studies were conducted using an MCR 302 (Premium Analytical Instruments, Anton Paar, Graz, Austria) rheometer equipped with an 8 mm parallel-plate torque measurement system using RheoCompass software. The tests were performed on 70 μl of liquid solutions at 1 mm gap size.

Viscosity-shear rates were measured over 1 to 100 s$^{-1}$ at room temperature at 0.5 mm gap size. These tests on GelCA-PEDOT composites were performed at 50 °C. For oscillatory experiments, 1 Hz frequency and 0.5% strain amplitude were defined unless noted otherwise. In order to assess the diffusive crosslinking of hydrogels, pre-gel solutions were submerged in crosslinking solutions, and storage modulus (G′) and loss modulus (G″) response were monitored through time sweep tests. The gelation point was defined as the crossover of G′ and G″ curves. To assess thermosensitivity, pre-gels were allowed to equilibrate at 50 °C for 5 min. Subsequently, a linear thermal transition from 50 °C to 10 °C at a rate of 1 °C/min was applied while G′ and G″ were recorded at 10% strain amplitude. Thermosensitivity tests were performed for PEDOT polymers at their dispersibility concentration limits (i.e., 20% w/v for PEDOT:AlgS and 4% w/v for PEDOT:PSS) incorporated in 12% w/v GelCA solutions.

## Mechanical tensile tests

Tensile mechanical properties of crosslinked GelCA-based conductive bioadhesives were evaluated using an Instron 5943 (MA, USA) universal testing equipment using BlueHill 3 software. To prepare the samples, 200 μl solutions were pipetted into 25 × 5 mm rectangular polydimethylsiloxane (PDMS) molds and crosslinked diffusively by 5 min soaking in 25 mM FeCl$_3$ solution. Then, crosslinked samples were glued to plain paper using Krazy glue (OH, USA) for better gripping. After loading the tests into the system, the tests were carried out at a crosshead displacement rate of 10 mm min$^{-1}$. The force-displacement data was recorded using a 100 N load cell. The failure point was defined as the first drop observed in the stress-strain curves, from which the tensile strength and strain at failure were determined.

## Ex vivo burst pressure tests

To evaluate the sealing effectiveness of the bioadhesive hydrogels, ex vivo burst pressure tests were conducted on porcine lung tissues. An oxygen reservoir was connected to a pressure gauge and a porcine lung through a T-shaped connector. A 3 mm wide puncture was thoroughly sealed using 100 μl bioadhesive pre-polymer (containing 12% w/v GelCA and certain concentrations of PEDOT:dopants) followed by 5 min diffusive crosslinking using 25 mM FeCl$_3$. Pressure changes were monitored as oxygen gas was introduced into the lungs until the point where the corresponding pressure was registered as burst pressure.

## Electrical measurements

The DC electrical conductivity was determined using a standard two-point probe setup via a Reference 600™ (Gamry Instruments, PA, USA) potentiostat. For conductivity of dry PEDOT:PSS and PEDOT:AlgS films, 75 μl samples taken from the PEDOT polymerization reactions were taken before the dialysis step at different time points and pipetted onto a glass slide, followed by dry annealing in an 80 °C oven. The dried coatings were cut into rectangular shapes, and then copper tape electrodes were affixed onto the coatings using silver glue, resulting in ~3 × 7 mm sample sizes. The coating thicknesses were measured to be ~4 μm using atomic force microscopy, based on which conductivity was calculated according to Eq. (2):

$$\sigma = \frac{tl}{Rw} \qquad (2)$$

Here, resistance (R) was calculated based on the average current output in response to 1 V applied voltage measured over 30 s, and t, l, and w are film thickness, length, and width, respectively. The conductivity of hydrogel samples was measured by bridging two gold electrodes placed at a 7 mm distance via injecting pre-polymer solutions on glass substrates. The applied hydrogel width and thickness

were ca. 5 mm and 1 mm, respectively, and the DC conductivity measurements were performed following the same procedure for dry coatings.

Impedance spectroscopy characteristics were assessed via a two-point probe setup using an AC voltage of $V_{RMS} = 100$ mV over 100 MHz to 1 MHz frequency range. Gamry Echem Analyst software (version 6.3) was used for fitting the circuit models to the impedance data.

## In vitro biocompatibility

The biocompatibility of the synthesized conductive polymers in GelCA matrices was evaluated through a 2D cell culture experiment using human dermal fibroblast cell lines (ATCC PCS201012). First, a pre-gel hydrogel mixture (100 µl) was introduced into PDMS molds with an 8 mm diameter and a 1.5 mm depth to create hydrogel disks. These hydrogel disks were subsequently immersed in a 25 mM $FeCl_3$ solution for 5 min for crosslinking. Afterward, the disks were immersed in a DPBS solution and sterilized under UV light for 1 h.

The cells were cultured and incubated at 37 °C with 5% $CO_2$, seeding them in a 24-well plate at a cell density of 2500 cells per well for 24 h in a 2 ml medium consisting of Dulbecco's modified Eagle's medium (DMEM) supplemented with 10% fetal bovine serum (FBS) and 1% penicillin-streptomycin. Subsequently, sterilized hydrogels were added to transwells (4 µm mesh), and the transwells were placed in the well plates containing the cells. Wells without hydrogels were used as a control.

To assess the metabolic activity of the cells, a PrestoBlue assay (ThermoFisher Scientific, USA) was performed at various time intervals. The cells were washed with DPBS and then incubated with 1 ml of PrestoBlue reagent (diluted 10× in the cultured medium) for 1.5 h at 37 °C. Subsequently, the PrestoBlue (100 µl per well) was transferred to a fresh 96-well plate, and fluorescence values were recorded at excitation/emission wavelengths of 530/590 nm using a microplate reader (BioTek Synergy 2, USA).

A Live/Dead assay (Invitrogen, USA) was conducted to assess cell viability. The cells were rinsed multiple times before applying the Live/Dead solution. The Live/Dead solutions were prepared by mixing 5 µl of calcein and 20 µl of ethidium homodimer-1 with 10 ml DPBS, and 500 µl of this solution was added to each well and incubated at 37 °C for 15 min. The cells were then rinsed with DPBS and imaged using a Zeiss fluorescence microscope (Axio Observer 5) with excitation/emission wavelengths of 494/515 nm for calcein (green) and 528/617 nm for ethidium homodimer-1 (red). ImageJ software was used to count live and dead cells, enabling the calculation of cell viability as the ratio of live cells to the total cell count.

## In vitro immunoactivity

The in vitro immunoactivity of PEDOT:AlgS was evaluated using bone marrow-derived macrophages (BMDMs), isolated from the tibia and femur bones of 5- to 6-week-old C57BL/6 mice. Approximately $10^6$ cells per dish were isolated and seeded in a mixture of DMEM supplemented with 10% w/v fetal bovine serum, 55 µM β-mercaptoethanol, 5 ng ml$^{-1}$ of macrophage colony-stimulating factor (MCSF), and 100 U/ml penicillin. After 8 days of culture, the BMDMs were seeded in a 6-well transwell plate at a density of $1 \times 10^6$ cells per well. Following overnight incubation, the macrophages were co-cultured with cross-linked disk-shaped PEDOT:AlgS hydrogels in a transwell plate, each having a volume of 100 µl hydrogel material, for 24 h. Subsequently, the cells were collected, and flow cytometry (Attune, ThermoFisher Scientific, USA) was used to assess polarization and activation for the stained macrophages. In a parallel study for microscopy analysis, the following steps were performed: After a 24-h incubation period, the cells were washed three times with PBS to remove any residual media or debris. Subsequently, the cells were fixed with 4% paraformaldehyde for 20 min to preserve their structure and morphology. To prevent non-specific binding of antibodies, the cells were then blocked

with 2% bovine serum albumin (BSA) for 90 min. Following the blocking step, the cells were incubated overnight at 4 °C with primary antibodies against F4/80 (a macrophage-specific marker) and CD80 (a co-stimulatory molecule) at a dilution of 1:250. After the overnight incubation, the cells were washed three times with PBS to remove any unbound antibodies. Subsequently, the cells were stained with DAPI (4′,6-diamidino-2-phenylindole) at a dilution of 1:1000 for 15 min. Finally, microscopy analysis was performed to visualize and study the stained cells.

## In vivo biocompatibility

Biocompatibility of the crosslinked PEDOT:AlgS was assessed in vivo by implantation in mice following an approved protocol at the Lundquist Institute (#22747-01). For each sample, five black male C57BL/6 mice, 6–8 weeks old, obtained from The Jackson Laboratory in the USA, were housed in standard laboratory conditions with a 12-h light/dark cycle, ambient temperature maintained at $25 \pm 2$ °C, and relative humidity of $50 \pm 10$%. Animals were provided ad libitum access to laboratory pellets and purified water in pathogen-free facilities. The PEDOT solutions were then injected (100 µl) intradermally. All surgeries were performed under anesthesia, with 1.5% v/v isoflurane in oxygen gas followed by carprofen injection for pain management. Tissues from the mice were subsequently harvested after 1 week following euthanasia via carbon dioxide inhalation. Subsequently, the samples were preserved by immersion in a 10% v/v formalin solution. Then, the preserved organs and tissues were encased in paraffin and cut into 5 µm-thick sections. The cellular morphology was assessed through hematoxylin and eosin (H&E) staining.

## In vivo immunofluorescence staining

For the subcutaneous implantation samples, immunofluorescent staining was performed. Anti-mouse F4/80 (BioRad, catalog number: MCA497A488, lot #18567, clone #Cl:A3-1), CD80 (BioLegend, catalog number: cat 600055, lot #b444189, clone W17200C), CD3 (BioLegend, catalog number: cat 100205, lot #B424757, clone 17A2), and Ly6G (BioLegend, catalog number: 127607, lot #B440678, clone 1A8), as well-established marker for identifying mouse macrophages, their pro-inflammatory subtypes, T cells, and neutrophils were used. For immunofluorescent staining, the slides were submerged in the xylene substitute for 5 min to deparaffinize the tissue sections. The tissues were rehydrated by washing them in ethanol (100%, 90%, and 70%, respectively) for 5 min each, followed by a 5-min wash in PBS. The slides were then submerged in an antigen retrieval buffer diluted 1:50 in MilliQ water, placed in a steamer set to 98.5 °C for 15 min, and subsequently incubated at room temperature for an additional 10 min to achieve antigen retrieval. Afterward, we circled the tissues using a hydrophobic pen. The slides were then washed twice in distilled water for 2 min each, followed by a 5 min wash in PBS. After drying, each tissue was covered with 10% normal goat serum for 30 min, placed in a humid chamber, and incubated in primary antibody solution overnight at 4 °C. From this stage onwards, the slides were submerged twice in PBS for 5 min each during the washing process. Subsequently, the slides were counterstained with DAPI, diluted at a ratio of 1:100 in PBS, for 10 min. Following this, the slides underwent a 10-min staining process with Sudan Black to diminish autofluorescence. Subsequently, the slides were coverslipped using an aqueous mounting medium and imaged using a Keyence microscope. Widefield fluorescent images were acquired post subcutaneous implantation, with color-coding as follows: Green for F4/80 macrophages, red for CD80, CD3, and Ly6G, and blue for DAPI.

## In vitro blood clotting assay

The blood clotting tendency of the material was measured in vitro with whole human blood containing 3.8% w/v sodium citrate anticoagulant. Two hundred µl of GelCA-based pre-gel mixtures were crosslinked in a

48-well plate using 25 mM FeCl₃. To initiate the clotting process, we thoroughly mixed 0.1 M calcium chloride with the blood at a 1:10 ratio and agitated the mixture for 10 s. Two hundred µl of the mixture was dispensed into the wells containing hydrogel, and the same amount was added to the empty wells as a control. At each designated time interval, the wells were rinsed with saline, ensuring the removal of soluble blood components until only the clots remained. The moment at which a consistent clot was formed in the wells was recorded as the clotting time. To assess the clot-forming abilities of different hydrogels, 10 ml of Milli-Q water was introduced to each sample after 16 min of clotting time, Afterward, the supernatant was analyzed using a plate reader at 405 nm. Greater clot formation was associated with lower absorbance values.

### Three-dimensional printing

The conductive inks comprising 20% w/v PEDOT:AlgS and 3% w/v alginate were 3D-printed on glass slide substrates (unless noted otherwise) using a three-axis robotic deposition stage (Aerotech). AutoCAD software was used to design the printing paths, and G-code was generated using a custom Python script (available at https://github.com/hmontazerian/DXF-to-GCode.git). The ink was inserted into 15 ml syringe barrels and equipped with 27 GA tapered tips. Extrusion of the ink was regulated by applying the air pressure using an EFD Nordson benchtop fluid dispenser. Before printing the patterns, the printing pressure and speed were optimized to be 5 psi and 2 mm s⁻¹ for stable extrusion.

### In vitro pH sensing tests

A screen-printed gold electrode (Aux.:Au; Ref.:Ag)/Ink AT as a three-electrode system and Autolab with PGSTAT10 potentiostat/galvanostat from Metrohm, USA, were used for pH sensing experiments. The gold screen-printed electrodes (SPEs) were rinsed with ethanol, followed by distilled (DI) water, and then dried under nitrogen gas flow. The hydrogel-modified pH sensing system was prepared by applying 10 µl of the GelCA-based bioadhesive pre-polymers on the gold working electrodes (WEs). The pH was controlled using buffer solutions at pH 4, 7, and 9 (Fisher Scientific, USA). To ensure a consistent pH level during chronoamperometry, we meticulously controlled the environmental factors by maintaining a constant temperature throughout the experiment. The chronoamperometry studies were performed at 0.65 V.

### In vivo pH sensing tests

The animal experiment was performed using ob/ob mice, B6.Cg-Lepob/J mice (The Jackson Laboratory, Bar Harbor, ME, USA) according to the protocol approved by the Institutional Animal Care and Use Committee (protocol no. IA23-1800) at California Institute of Technology. After anesthesia and analgesia, 5-mm full-thickness wounds were created using a blade, and biosensors were implanted in subcutaneous pockets along the dorsomedial skin. *S. aureus* and *E. coli* were used for antimicrobial tests. After that, the electrodes were applied to the wound directly and tested using the hydrogel electrodes. The animals were tested for a continuous 3-day period. A mixed bacteria solution with a concentration of $1 \times 10^8$ CFU of each bacteria species was added after the first day of the test, and TCP-25 (GKYGFYTHVFRLKKWIQK-VIDQFGE) (98% purity, acetate salt) (CPC Scientific) was applied after the secondary day of the test. Current measurement was performed at 0.1 V and the average current measured over 30 s was reported for each measurement time increment.

### Antibacterial tests

The antimicrobial activity of hydrogels was evaluated using an agar well diffusion assay. *E. coli* microbial strains were cultured in nutrient broth until reaching an optical density (OD600) of 1.24 for *E. coli* measured with a UV-Vis spectrometer. Agar plates were prepared by spreading the inoculum over the surface. Pre-gel solutions (100 µl) were pipetted on the agar plates. Crosslinked hydrogels were placed directly on the agar surface. The plates were incubated at 37 °C for 24 h to observe the zones of inhibition. As a positive control, 8 mm filter paper disks impregnated with 15 µg of silver sulfadiazine were placed on the agar plates.

### In vivo wound healing

Circular full-thickness wounds were created on the dorsal skin of the animal model using an 8 mm biopsy punch. After wound creation, the wounds were treated with the designated hydrogels and covered as needed to maintain the treatment. Wound area measurements were taken on days 6 and 12 post-treatment using images captured to track the healing process.

### Biopotential signals recording

The bioadhesive GelCA-based conductive solutions were used to record biopotential signals. The reference electrode was placed on the leg for both electrocardiogram (ECG) and electromyography (EMG) monitoring. Working electrodes were placed on the left and right arms for ECG monitoring. While for EMG monitoring, the working electrodes were placed at both ends of the biceps brachii muscle. For both biopotential signals monitoring, the signal was acquired through an open-source hardware shield (SparkFun, AD8232) and was processed by a digital lowpass filter (50 Hz) through a MATLAB code (available at https://github.com/hmontazerian/biopotential-signals-recording.git).

### Temperature sensitivity tests

The temperature-sensing characterization was performed on a ceramic hotplate (ThermoFisher Scientific) where the temperature was monitored using a thermometer. A two-probe setup using an amperometric method was used through an electrochemical workstation (CHI 660E) to record resistance changes with temperature under an applied voltage of 1 V.

### Statistical analyses

The reported values are presented as means ± standard deviation (SD). GraphPad Prism 10 software was employed for statistical analyses using analysis of variance (ANOVA), with statistical significance considered when $p < 0.05$. Normal data distribution was assumed for parametric tests. All the experimentally measured data were conducted in triplicate ($n = 3$) for independent samples unless otherwise specified.

### Ethics

Every experiment involving animals, human participants, or clinical samples has been carried out following a protocol approved by an ethical commission.

### Reporting summary

Further information on research design is available in the Nature Portfolio Reporting Summary linked to this article.

## Data availability

All data supporting the findings of this study are available within the article and its supplementary files. Any additional requests for information can be directed to and will be fulfilled by the corresponding authors. Source data are provided with this paper.

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

## Acknowledgements

The authors acknowledge the financial support from the National Institutes of Health (R01EB023052, R01HL140618, R01EB031992, R01HL155815, and R01DC021461) and Heritage Medical Research Institute. E.D. thanks for the support from the National Institutes of Health Training Grant (T32EB023858).

## Author contributions

The original idea was developed, and experiments were designed by H.M. and E.D. H.M. prepared the majority of the original draft and performed data analysis in addition to conducting electrical characterizations, rheological studies of viscosity and thermal gelation, DLS, and testing the degree of sulfonation. E.D. led the dispersibility tests, rheological experiments of ionic crosslinking, UV-vis experiments, as well as 3D printing of hydrogels. F.L. conducted pH testing of conductive bioadhesives in vitro, and C.W. assessed the wound monitoring capability of hydrogels in vivo. In vitro biocompatibility tests were performed by R.H. R.R.S. contributed to drafting the experimental section and assisted H.M. in sulfonation degree measurements. J.L. performed temperature sensitivity, physiological monitoring studies and assisted E.D. with 3D printing experiments. S.K. performed mechanical characterizations. The hemostatic analyses were led by F.Z. The AFM imaging and FTIR tests were performed by T.L. and N.H., respectively. Y.Z. assisted H.M. in conducting electrical characterizations. In addition to in vivo biocompatibility, A.H.N. conducted in vitro immunoactivity studies, SEC experiments for biodegradability, and PEDOT polymerization. N.M. assisted A.H.N. in flow cytometric studies of immunoactivity. T.H., R.S.L., D.G.A., A.K., W.G., and P.S.W. supervised, conceived of this study, and contributed to the writing of the manuscript. The final version of the manuscript has been approved by all authors.

## Competing interests

The authors declare no competing interests.

## Additional information

[1]David H. Koch Institute for Integrative Cancer Research, Massachusetts Institute of Technology, Cambridge, Massachusetts, USA. [2]Department of Bioengineering, University of California, Los Angeles, Los Angeles, California, USA. [3]Mechanical Engineering Department, University of Utah, Salt Lake City, Utah, USA. [4]Terasaki Institute for Biomedical Innovation, Los Angeles, California, USA. [5]Andrew and Peggy Cherng Department of Medical Engineering, Division of Engineering and Applied Science, California Institute of Technology, Pasadena, California, USA. [6]Department of Engineering Science and Mechanics, Pennsylvania State University, University Park, Pennsylvania, USA. [7]Department of Chemistry and Biochemistry, University of California, Los Angeles, Los Angeles, California, USA. [8]Department of Electrical and Computer Engineering, University of British Columbia, Vancouver, British

Columbia, Canada. ⁹Department of Chemical Engineering, Massachusetts Institute of Technology, Cambridge, Massachusetts, USA. ¹⁰Department of Anesthesiology, Boston Children's Hospital, Boston, Massachusetts, USA. ¹¹Institute for Medical Engineering and Science, Massachusetts Institute of Technology, Cambridge, Massachusetts, USA. ¹²Harvard-Massachusetts Institute of Technology Division of Health Sciences and Technology, Massachusetts Institute of Technology, Cambridge, Massachusetts, USA. ¹³Department of Materials Science and Engineering, University of California, Los Angeles, Los Angeles, California, USA. ¹⁴These authors contributed equally: Hossein Montazerian, Elham Davoodi. ✉e-mail: hassania@terasaki.org; psw@cnsi.ucla.edu; khademh@terasaki.org; weigao@caltech.edu

