## [Transparent Peer Review file · Nature Communications]

Boosting hydrogel conductivity via water dispersible conducting polymers for injectable bioelectronics

Corresponding Author: Professor Wei Gao

Version 0:

Reviewer comments:

Reviewer #1

(Remarks to the Author)

The authors proposed a novel approach to develop an injectable conductive bioadhesive that addresses the unmet needs of conventional PEDOT in implantable bioelectronics. This approach enhances the dispersibility of the conductive ink many-fold, allowing for higher PEDOT packing/percolation, and incorporates the matrix with an adhesive polymer. Thus, the reviewer recommends acceptance into the journal after revisions to address the following issues:

1. In the introduction, the following recent references regarding injectable conductive bioadhesives should be cited:

-Jin et al., *Nature* 2023, 623, 58–65

- Park et al., *Small* 2023, 2300250

- Hsieh et al., *Device* 2024, 2, 100182

- Kim et al., *Advanced Materials* 2023, 2307070

2. How does PEDOT manufactured at 0.9 w/v% compare to commercially available PEDOT, such as Clevios PH1000, which is a commonly compared PEDOT product? The dispersion properties of AlgS likely allow for a low-viscosity solution with higher EDOT concentrations to be added and polymerized compared to a 1:2.5 PEDOT ratio, resulting in more PEDOT along with AlgS compared to PSS. These advantages should be discussed in detail.

3. As mentioned by the authors, PEDOT generally does not degrade in the body. In contrast, PEDOT appears to undergo degradation. An explanation of the reasons and mechanisms behind this difference is necessary. It is important to clarify that the degradation of AlgS via hydrolysis does not imply the degradation of PEDOT as a whole. Does PEDOT itself remain undegraded, or does it also degrade or erode as a result of AlgS degradation?

4. The author used ferric chloride to prepare PEDOT-based hydrogels. Does the use of ferric chloride affect the final hydrogel's electrical properties? It is noted that ferric ions are not present in PEDOT but are present in PEDOT. This difference could potentially have a significant impact on the material's properties and electrical characteristics.

5. Figure 2b shows a characteristic peak at 700–900 nm for the polaron state of PEDOT, not necessarily the bipolaron state. The authors are advised to further analyze the NIR range (900–1200 nm) of the PEDOT using UV-Vis-NIR spectroscopy to reveal the characteristic peaks of the PEDOT bipolaron state. Additionally, relative absorbance with baseline correction for Fig. 2b and Supplementary Fig. 3a is recommended.

6. Supplementary Figure 1d indicates a 35% degree of sulfonation at 1.5% w/v CSA, while a 70% degree of sulfonation is mentioned in the main text. Elaborate on why the degree of sulfonation is doubled.

7. Please provide phase-angle EIS data for the GelCA-PEDOT composite.

8. For Supplementary Figure 20, is the incubation time 24 hours as indicated in the scheme or 1 hour as indicated in the figure caption?

9. Please remove the 'e' typo on Page 5, "... From a microstructural perspective, e scanning electron microscopy ...".

Reviewer #2

(Remarks to the Author)

The manuscript presents a novel type of soft wearable electronics featuring PEDOT:AlgS with enhanced aqueous dispersibility and conductivity, integrated into various hydrogel systems. This study substantiates the design strategy that hydrophilic dopants can effectively modify the multiple properties of conducting polymers, leading to a broad spectrum of applications in biomonitoring. First, the conductive PEDOT-AlgS is hydrophilic, biocompatible, degradable, and can be

easily prepared via 3D printing. Second, these additives are compatible with various hydrogel systems to construct wearable electronics with excellent physiological monitoring functions. These properties position PEDOT-AlgS as a promising candidate for implantable, biodegradable, and injectable bioelectronics. The manuscript is well-written, well-designed, and easy to follow. Below are some comments for the manuscript:

1. In the introduction section, the authors should explain why sulfonated alginate was chosen as the dopant for PEDOT.
2. Hydrophilicity is one of the advantages of PEDOT:AlgS compared to conventional conductive polymers. Could the authors provide statistics on the water contact angles of both groups?
3. Biocompatibility and biodegradability are two crucial challenges in implantable bioelectronics. The authors attributed the degradability of PEDOT:AlgS to the hydrolysis of glycosidic bonds in alginate. Is PEDOT:AlgS degradable in an in-vitro and cell-free environment, such as PBS or a cell culture medium at 37°C? What is the complete degradation time both in vivo and in vitro? What are the primary degradation products and their potential metabolic pathways? Please provide more detailed information.
4. In Figure 3, the Hematoxylin and eosin staining of intradermally injected solutions of PEDOT:AlgS and PEDOT:PSS showed that the PEDOT:AlgS group exhibits a loss of follicles and accumulation of fat tissues. Is this phenomenon due to the different implantation sites or the consequence of the different polymer components?
5. In the results section, the authors stated that "In vivo injection of 5% w/v PEDOT:PSS solutions resulted in the formation of a fibrous capsule around PEDOT:PSS, whereas immune cells infiltrated PEDOT:AlgS samples, attempting to digest the PEDOT polymers (Figure 3h)." It would be beneficial to identify the specific types of immune cells involved in this process. Further immune staining or flow cytometry (FACS) analysis is recommended to provide a more detailed characterization of the immune response to PEDOT:AlgS. Sequencing analysis might be valuable.
6. In Supplementary Figure 9, the immunostaining of F4/80 is unclear. Please provide higher magnification images for better visualization. Additionally, including statistical analysis of the F4/80 staining results would enhance the clarity and impact of the findings.
7. The pH sensing property is a significant highlight of the GelCA-PEDOT-AlgS bioadhesive application. The authors attributed the pH sensing mechanism in GelCA-PEDOT to catechol-to-quinone transitions by CA. This assumption should be supported by appropriate compositional analysis. Moreover, as catechol is easily oxidized in air, the pH sensing accuracy and sustainability are questionable. The authors should test the pH sensing accuracy at different time points after implantation to indicate the validity period of the bioelectronic device.
8. A major concern in the application of hydrogel-based wound monitors is the risk of infection, which can hinder effective treatment. Therefore, the hydrogel is expected to possess anti-infective and wound healing properties. It is crucial to evaluate whether PEDOT-AlgS, either alone or in a hydrogel system, can exert a therapeutic effect on infectious wound healing. Please examine the effect of PEDOT-AlgS on skin healing status and its anti-infective properties.
9. In the Methods section, please specify the sample size, sex, age, and ethical approval of the animals used in the study.
10. Please consider expanding the discussion on the potential future applications of injectable conductive hydrogel monitoring devices. Including a dedicated section in the conclusion or discussion that outlines the prospects, challenges, and possible advancements in this field would provide valuable insights for readers and highlight the broader impact of your work.

Reviewer #3

(Remarks to the Author)

The manuscript authored by Wei Gao and co-workers reports the synthesis of conductive hydrogels made of PEDOT doped with sulfonated alginate (AlgS) to be used as functional electrodes for soft injectable electronics. The idea is interesting, but there are several errors (Error! Reference source not found) that make it impossible to read the manuscript and relate the text with the figures for a correct evaluation of the work. Therefore, the authors need to solve these issues and resubmit the manuscript so that it can be reviewed and evaluated.

Version 1:

Reviewer comments:

Reviewer #1

(Remarks to the Author)

The authors have revised their manuscript to address clearly the issues raised by the reviewer. The reviewer now recommends the manuscript for publication in this journal.

Reviewer #2

(Remarks to the Author)

The authors have effectively addressed the concerns raised by providing solid evidence. Therefore, I recommend the acceptance of this manuscript.

Reviewer #3

(Remarks to the Author)

This manuscript reports the synthesis of hydrophilic PEDOT dispersions by doping this conducting polymer with sulfonated alginate (AlgS) instead of PSS. This increased the conductivity of the PEDOT:AlgS dispersions compared to commercial

PEDOT:PSS. The authors carried out an in-depth study of the properties of this dispersion, as well as of its use in electrically functionalized adhesive gelatin hydrogels for biosensing purposes. Although the idea presented here is interesting and the work is well conducted, the following points need to be clarified to improve the quality of the manuscript.

- 1) On page 5 it is mentioned that “While sulfonation interrupted alginate ionic associations, possibly due to steric hindrance and conformation changes, comparable mechanical properties could be achieved by increasing AlgS content, leveraging its enhanced solubility.” Therefore, why was the elastic modulus of AlgS2 hydrogels lower than that of AlgS1 hydrogels?
- 2) I suggest calling “sulfonated alginate-doped PEDOT polymers” instead of “PEDOT-doped sulfonated alginate polymers”
- 3) I suggest showing the whole wavelength range (225 – 1200 nm) for all UV-vis spectra to establish a clear comparison of the different conditions tested in the same range (Figure 2a and Supplementary Figure 3a-c).
- 4) What was the conductivity of PSS-doped PEDOT after 2 d (Supplementary Figure 4)?
- 5) How were the authors able to distinguish between PEDOT and AlgS in the SEM images (Figure 2e and Supplementary Figure 6)? A chemical scanning technique must be used to distinguish between two different chemical components in the topography (for example, nanoFTIR).
- 6) The molecular weight of the sodium alginate used needs to be determined, as it may explain the differences observed with PSS (70,000 Da).
- 7) What happens to the mean hydrodynamic diameter of PEDOT:PSS at different concentrations after freeze-drying (Supplementary Figure 7a)?
- 8) What is the meaning of the inverted eppendorf shown in Figure 3e?
- 9) To prove that PEDOT:AlgS is ionically responsive to Fe³⁺ (Figure 3g), the evolution of the elastic modulus of the PEDOT:AlgS2 dispersion over time without contact with FeCl₃ should be shown.
- 10) How long did the hydrolytic degradation take to obtain the results shown in Figure 3i and Supplementary Figure 10? The differences between the spectra before and after hydrolysis are not large.
- 11) On page 8, it is mentioned that coatings were formed with the 2.5% w/v solutions) (Figure 3jll and Supplementary Figure 11c–e), but in the caption of Supplementary Figure 11c–e it is written that AFM imaging of coatings obtained from 5% w/v PEDOT solutions. The same concentration should be used in all cases to establish a comparison between samples.
- 12) What about the comparison of the conductivity properties of the current work (Supplementary Figure 12) with those of reference 25 of the main draft (Yuk H, Lu B, Lin S, Qu K, Xu J, Luo J, et al. 3D printing of conducting polymers. Nat. Commun. 11, 1604 (2020)?
- 13) On page 9, it is mentioned that in terms of printing fidelity (Figure 4e), solutions of 4% w/v PEDOT:PSS in alginate experienced multiple clogging events due to aggregation and precipitation, but these results are not shown in Figure 4e. I suggest including the results concerning to 3D printing tests of PEDOT:PSS + alginate in the supporting information file for comparison with those of PEDOT:AlgS + alginate shown in Figure 4e.
- 14) On page 10, it is written that “The potential amplitudes for T waves of PEDOT:AlgS were found to be ~30% greater than PEDOT:PSS”, but this is not visible in Figure 4h. They show similar amplitudes.
- 15) The control used for cell tests shown in Supplementary Figure 16 needs to be specified.
- 16) What the authors observed in Supplementary Figure 17 is not a sol-gel transition with the temperature, but the exact opposite, they observed a gel-sol transition. For example, at temperatures from 10 °C to 29 °C for GelCA, G' is higher than G'', which is the condition for the gel state; however, from 29 °C, G'' is higher than G', which is the conditions for the solution (sol) state. Therefore, the main text and the caption of this figure must be corrected accordingly.
- 17) How did the authors determine the tensile strain at failure (Supplementary Figure 18d)? I suppose those values were obtained from stress-strain curves shown in Supplementary Figure 18a, and I do not understand why there is no difference between GelCA+PEDOT:PSS and GelCA+PEDOT:AlgS for tensile at failure when there are large differences in the curves of the Supplementary Figure 18a.
- 18) The results of the clotting time corresponding to the blank control should be included in the bar plot in Figure 5f.
- 19) What are the differences between the two bars of the plot included in the Supplementary Figure 22e? Both bars show the same sample name.
- 20) On page 12, it is mentioned that “The pH sensing mechanism in GelCA-PEDOT is attributed to synergistic effects of mediated electron and ion mobility due to hydrogel swelling”. To prove this, swelling tests should be performed and the results should be included in the supporting information file.
- 21) The caption of Figure 5 needs to be revised as the label (E) should be replaced by e.

Version 2:

Reviewer comments:

Reviewer #3

(Remarks to the Author)

After reviewing the responses and corrections made by the authors, the article can be accepted for publication in its current form.

Responses to the reviewers' comments

We thank all three reviewers for taking time to read our manuscript and providing very constructive and valuable feedback. We have revised the manuscript substantially as per the reviewers' suggestions. The revised text is highlighted in yellow in the main text and below is a response to each comment.

Reviewer #1:

General Comment. The authors proposed a novel approach to develop an injectable conductive bioadhesive that addresses the unmet needs of conventional PEDOT in implantable bioelectronics. This approach enhances the dispersibility of the conductive ink many-fold, allowing for higher PEDOT packing/percolation, and incorporates the matrix with an adhesive polymer. Thus, the reviewer recommends acceptance into the journal after revisions to address the following issues:

Response: We thank the reviewer for very positive feedback for our manuscript and the insightful comments. We have revised the manuscripts by addressing the suggestions noted by the reviewer as detailed below. We believe that these changes have substantially improved our manuscript.

Comment 1. In the introduction, the following recent references regarding injectable conductive bioadhesives should be cited:

- Jin et al., Nature 2023, 623, 58–65
- Park et al., Small 2023, 2300250
- Hsieh et al., Device 2024, 2, 100182
- Kim et al., Advanced Materials 2023, 2307070

Response: We thank the reviewer for the positive and constructive feedback. We have reviewed the recommended references and agree that they are relevant to our manuscript. We have now included citations to these papers in the introduction.

Changes (p. 3): The suggested references^{R1-R4} have been added to the introduction section.

“These networks can be engineered further to introduce various functionalities including tissue regenerative effects, stimuli-responsiveness, bioadhesion, and more¹³⁻¹⁶.”

Comment 2. How does PEDOT manufactured at 0.9 w/v% compare to commercially available PEDOT, such as Clevios PH1000, which is a commonly compared PEDOT product? The dispersion properties of AlgS likely allow for a low-viscosity solution with higher EDOT concentrations to be added and polymerized compared to a 1:2.5 PEDOT ratio, resulting in more PEDOT along with AlgS compared to PSS. These advantages should be discussed in detail.

Response: The PEDOT-to-dopant ratio synthesized at 0.9% w/v here is 0.9:1, which is >2× larger than this ratio for the commercially available Clevios PH1000, *i.e.*, 1:2.5. Therefore, instead of using commercial Clevios PH1000 as a control, we synthesized PEDOT:PSS in-house at the same

PEDOT:dopant ratio as PEDOT:AlgS (*i.e.*, 0.9:1) to elucidate the effects of AlgS in enhanced dispersion PEDOT, particularly at this high PEDOT:dopant ratio where dispersion is challenging.

As shown in Figure R1, in Supplementary Figure 3f we presented viscosity-shear rate profiles for the 10 wt.% solutions of PEDOT:AlgS and PEDOT:PSS (synthesized at higher PEDOT:dopant ratio than commercial Clevis PH1000) suggested much lower viscosity for PEDOT:AlgS. It means that, as compared with PSS, AlgS dopants allow for lower-viscosity solutions at high EDOT concentrations thereby more PEDOT along with AlgS compared to PSS, which is well aligned with what the Reviewer suggests.

Figure R1. Viscosity-shear rate profiles for 10% w/v solutions of PEDOT:AlgS2 and PEDOT:PSS in water.

We performed further experiments to compare the rheological properties of PEDOT:AlgS with commercial PEDOT:PSS as per the Reviewer’s suggestion. We synthesized PEDOT:AlgS at a similar PEDOT:dopant ratio as Clevis PH1000 (1:2.5) and compared its viscosity-shear rate profiles (with similar solid content of 1.3% w/v) with Clevis PH1000, as shown in Figure R2. The results led to consistent outcomes where viscosity of commercial PEDOT:PSS led to approximately 15 times larger viscosity than PEDOT:AlgS. This result reiterated that AlgS likely allow for a low-viscosity solution with higher EDOT concentrations. We added these data to the revised manuscript and discussed them further for greater clarity.

Figure R2. Comparison of viscosity-shear rate curves of commercial PEDOT:PSS and synthesized PEDOT:AlgS prepared at a similar PEDOT to dopant ratio (1:2.5) and dispersed at the same concentration in water (1.3 wt.%).

Changes (p. 7): We added more statements to the section “Aqueous re-dispersion of PEDOT-based conductive polymers” to clarify larger PEDOT polymerization capacity when using AlgS as dopants.

“This result not only indicates better dispersibility of PEDOT:AlgS, but also suggests that AlgS dopants allow for more PEDOT polymerization compared to PSS before reaching the dispersion limit, particularly at the PEDOT:dopant ratios exceeding those of commercially available Clevois PH1000.”

Furthermore, the following discussions were added to the section “Design and characterization of water-dispersible PEDOT conductive additives” (p. 5):

“Overall, the DC and AC electrical characterizations suggest the formation of percolation networks at EDOT concentrations of greater than 0.5%, which is comparable to that of commercially available PEDOT (Clevios PH1000). A higher EDOT concentration of 0.9% w/v was used for subsequent characterizations to demonstrate the capability of AlgS dopants as a means to enable larger concentration loading of PEDOT in aqueous systems.”

In addition, the data shown in Figure R2 are now added to the Supplementary Figure S7f, as shown below, and related discussions were added to section “Aqueous re-dispersion of PEDOT-based conductive polymers”:

Supplementary Figure 7 | Characterization of PEDOT:AlgS re-dispersions in water after freeze-drying. **a**, Effects of sulfonation degree and EDOT monomer on the average hydrodynamic size results from DLS tests. **b**, Colloidal stability of PEDOT:AlgS measured in terms of the ratio of supernatant absorbance (after the 0.5% w/v solutions were left at rest for three months) to the absorbance of homogeneous solution of PEDOT:PSS and PEDOT:AlgS2 in water. **c**, Water contact angle of PEDOT:PSS and PEDOT:AlgS coatings air-dried on glass substrates. **d,e**,

Viscosity-shear rate characteristics of PEDOT:AlgS and PEDOT:PSS solutions, respectively, in water at different concentrations. **f**, Comparison of viscosity-shear rate curves of commercial PEDOT:PSS and synthesized PEDOT:AlgS prepared at a similar PEDOT to dopant ratio (1:2.5) and dispersed at the same concentration in water (1.3 wt.%).

“Similarly, commercial PEDOT:PSS solutions compared with PEDOT:AlgS synthesized at the same PEDOT to dopant ratio (1:2.5) resulted in ~15× larger viscosity in solutions of similar concentrations (1.3% w/v) as shown in Supplementary Figure 7f.”

We added “The commercial PEDOT:PSS (1.3 wt.% dispersion in H₂O) was supplied by Sigma Aldrich, USA.” to “Materials and reagents” section.

Comment 3. As mentioned by the authors, PEDOT generally does not degrade in the body. In contrast, PEDOT appears to undergo degradation. An explanation of the reasons and mechanisms behind this difference is necessary. It is important to clarify that the degradation of AlgS via hydrolysis does not imply the degradation of PEDOT as a whole. Does PEDOT itself remain undegraded, or does it also degrade or erode as a result of AlgS degradation?

Response: We show that PEDOT:AlgS offers better degradability as compared to PEDOT:PSS both at the chemical (molecular scale) and physical (larger macro-scale) levels when implanted *in vivo*. As the reviewer highlighted, better molecular degradability of PEDOT:AlgS arises from the dopant component, *i.e.*, AlgS due to hydrolyzable bonds in its chemical structure, which is absent in PSS. Our results do not suggest molecular degradation of the PEDOT phase. We clarified this point in the revised manuscript to avoid confusion. The macro-scale physical degradability of the PEDOT:AlgS solutions is also attributed to the lower viscosity of the PEDOT:AlgS complexes in aqueous medium and is also not related to PEDOT phase degradation.

Changes (p. 8): The following statement was added to the “Biocompatibility and biodegradability of PEDOT solutions” section:

“The enhanced molecular degradability of PEDOT:AlgS is attributed to the hydrolyzable glycosidic bonds on the backbone of AlgS, which are absent in the PSS structure (see Supplementary Figure 10). While PEDOT is generally stable, the byproducts of alginate are primarily alginate backbone broken into oligo- and monosaccharides involving sulfonated mannuronic acid (M) and guluronic acid (G) residues. The PEDOT phase is expected to remain intact due to the stable bonds, however, its smaller size distribution can promote their renal clearance in vivo.”

Comment 4. The author used ferric chloride to prepare PEDOT-based hydrogels. Does the use of ferric chloride affect the final hydrogel’s electrical properties? It is noted that ferric ions are not present in PEDOT but are present in PEDOT. This difference could potentially have a significant impact on the material's properties and electrical characteristics.

Response: Thank you for pointing out the effects of ionic crosslinking on electrical performance. We use FeCl_3 for the ionic crosslinking of the matrix phases in the PEDOT-incorporated hydrogels *via* diffusion by soaking the injected prepolymer in the crosslinking solution. To investigate the effects of FeCl_3 crosslinking, we compared the electrical resistance of GelCA hydrogels before and after diffusive crosslinking in FeCl_3 solution for 5 min and overnight as shown in Figure R3. The results indicate an improvement in conductivity due to enhanced ionic conduction *via* FeCl_3 which can drastically increase (up to $\sim 35\times$ matrix conductivity) *via* longer incubation in FeCl_3 . We note that the measurements could be underestimates due to diffusion gradients through the hydrogel thickness as electrodes in the conductivity test setup are located on the substrate where FeCl_3 concentration is the least. We expect this effect to be minimal in the long (overnight) crosslinking condition. We believe that there is a slight typographical error in the second part of Reviewer’s comment in “It is noted that ferric ions are not present in PEDOT but are present in PEDOT”, however, we added more statements to the revised manuscript to highlight the differences in material’s electrical properties due to crosslinking.

Figure R3. The maximum relative increase in conductivity of 12% w/v GelCA attainable with PEDOT:AlgS and PEDOT:PSS at their dispersibility limit before and after diffusive crosslinking in 25 mM FeCl_3 .

Changes (p. 11): The following statements were added to the revised manuscript (section “Hydrogel-based 3D printed soft bioelectrodes”):

“Crosslinking of PEDOT:AlgS hydrogels using FeCl_3 further increased this figure to $\sim 9.5\times$, which could be elevated up to over $35\times$ depending on the crosslinking time.”

The conductivity results shown in Figure R3 were incorporated in Supplementary Figure 19a as can be seen in the update figure below:

Supplementary Figure 19 | Electrical characterization of conductive GelCA-based composites comprising of PEDOT additives. **a**, The maximum relative increase in conductivity of 12% w/v GelCA attainable with PEDOT:AlgS and PEDOT:PSS at their dispersibility limit before and after diffusive crosslinking in 25 mM FeCl₃. **b**, **c**, Nyquist plots and phase Bode plots, respectively, obtained from impedance spectroscopy for GelCA solutions and its composites with synthesized PEDOT additives at their dispersibility limits. Z_{img} and Z_{real} are imaginary and real parts of impedance. **d**, The results of fitting the impedance data to the equivalent circuit model shown in the inset of **b**. R_i and R_p are resistances of their corresponding resistors in **b**, C_{dl} is the capacitive constant of capacitor C_{dl} , Q_p , and n_p are the constant and exponential values of the capacitive phase element (CEP_p).

Comment 5. Figure 2b shows a characteristic peak at 700–900 nm for the polaron state of PEDOT, not necessarily the bipolaron state. The authors are advised to further analyze the NIR range (900–1200 nm) of the PEDOT using UV-Vis-NIR spectroscopy to reveal the characteristic peaks of the PEDOT bipolaron state. Additionally, relative absorbance with baseline correction for Fig. 2b and Supplementary Fig. 3a is recommended.

Response: We appreciate the Reviewer for correcting us in our interpretation of PEDOT state with respect to their UV-vis characteristics. We have added UV-vis results for PEDOT:AlgS at wavelengths greater than 900 nm to support absorption bands over wavelengths associated with both polaron and bipolaron states as shown in Figure R4. We find absorption bands at wavelengths larger than 600 nm expanding through >800 nm wavelengths.

Figure R4. UV-vis spectra of PEDOT:AlgS at the near-infrared wavelength ranges.

According to Ref. R5, absorption bands of 600-900 nm and 700-1200 nm are attributed to polaron, and bipolaron states, respectively. The discussions related to the PEDOT state were updated accordingly and the labels in Figure 2a and Supplementary Figure 3a were corrected. Furthermore, the UV-vis data shown in Figures 2b and Supplementary Figure 3a were baseline corrected linearly and normalized by their peaks in the updated form shown in Figure R5a,b, respectively.

Figure R5. a, UV-vis spectra of PEDOT:AlgS2 at different polymerization times. EDOT, 3,4-ethylenedioxythiophene; APS, ammonium persulfate. The inset shows the absorbance of PEDOT:AlgS reaction solutions at 800 nm before freeze drying. **b,** UV-vis spectra of PEDOT:AlgS synthesized at different EDOT concentrations dispersed in water.

Changes (p. 6): The supplementary Figure 3 was updated where the PEDOT polaron label was corrected in panel a and Figure R4 was inserted as panel c as shown below:

Supplementary Figure 3 | Characterization of freeze-dried PEDOT:AlgS. **a**, UV-vis spectra of PEDOT:AlgS synthesized at different EDOT concentrations dispersed in water. The shoulder peak at wavelengths > 700 nm suggests that PEDOT stayed in doped state after dialysis and freeze-drying processes. **b**, Effects of EDOT concentration and alginate sulfonation degree on UV-vis spectra of PEDOT:AlgS. The magnitude of the absorbance was larger for higher sulfonation and EDOT concentrations. **c**, UV-vis spectra of PEDOT:AlgS at the near-infrared wavelength ranges. **d**, FTIR spectra of PEDOT:AlgS synthesized at different EDOT concentrations. **e**, Results of SEC characterizing molecular weight distribution of PEDOT:AlgS at varying EDOT concentrations.

The following statements were added to section “Design and characterization of water-dispersible PEDOT conductive additives” (p. 5):

“Near-infrared absorption bands at wavelength ranges of 600-900 nm, and 700-1200 nm suggest a transition to the polaronic and bipolaronic states, respectively, due to doping via anionic sulfonate groups of AlgS (Supplementary Figure 3c)³⁶. The results also indicate preservation of the doped state after dialysis and freeze drying.”

Figure 2 was updated as follows to reflect the updates in Figure 2b:

Figure 2 | Characterization of freeze-dried PEDOT-doped sulfonated alginate polymers. a, Polymerization and doping of PEDOT. The AlgS samples are labeled as AlgS x where x represents the % w/v concentration of CSA used in sulfonation reaction. **b,** UV-vis spectra of PEDOT:AlgS2 at different polymerization times. EDOT, 3,4-ethylenedioxythiophene; APS, ammonium persulfate. The inset shows the absorbance of PEDOT:AlgS reaction solutions at 800 nm before freeze drying. **c,** Effect of alginate sulfonation on the molecular weight distribution of freeze-dried PEDOT:AlgS2 tested *via* size-exclusion chromatography (SEC). **d,** Freeze-dried PEDOT forming large aggregates when doped with PSS while doping with AlgS2 results in homogeneously distributed small nanoparticles. **e,** Scanning electron microscopy (SEM) images of freeze-dried PEDOT:AlgS2 and PEDOT:PSS.

The sentence below was added to the Methods section (p. 16):

“UV-vis measurement in the near-infrared region was performed using an Agilent 8453 UV-vis Spectrometer.”

Comment 6. Supplementary Figure 1d indicates a 35% degree of sulfonation at 1.5% w/v CSA, while a 70% degree of sulfonation is mentioned in the main text. Elaborate on why the degree of sulfonation is doubled.

Response: We apologize for the confusion. This error was due to a miscalculation that was fixed in the figure prior to submission but was missed in the text. We have now fixed the degree of sulfonation stated in the revised manuscript.

Changes (p. 4): The abovementioned statement in section “Design and characterization of water-dispersible PEDOT conductive additives” was revised as follows:

“The sulfonation degree results (Supplementary Figure 1d) suggest approximately 37% conversion of hydroxyl groups in alginate when CSA concentration exceeded 1.5% w/v, reaching a plateau.”

Comment 7. Please provide phase-angle EIS data for the GelCA-PEDOT composite.

Response: We have now added the phase-angle EIS data for GelCA-PEDOT composites as shown in Figure R6. We observed that the addition of PEDOT to GelCA led to phase angle drops at higher frequencies than GelCA alone and recorded overall decreasing trends for both PEDOT:PSS and PEDOT:AlgS groups.

Figure R6. Phase Bode angle-frequency plots for GelCA and its composites with synthesized PEDOT additives at their dispersibility limits.

Changes (p. 11): Supplementary Figure 19 has been updated to include Figure R6 (see panel c) and the main text was updated in section “Injectable smart bioadhesives for wound monitoring”:

“The EIS results for PEDOT-incorporated GelCA solutions, along with the equivalent circuit constants (Figure 5d and Supplementary Figure 19b-d), corroborate the greater capacity of PEDOT-AlgS additives to impart conductivity to GelCA hydrogels.”

Supplementary Figure 19 | Electrical characterization of conductive GelCA-based composites comprising of PEDOT additives. **a**, The maximum relative increase in conductivity of 12% w/v GelCA attainable with PEDOT:AlgS and PEDOT:PSS at their dispersibility limit before and after diffusive crosslinking in 25 mM FeCl₃. **b**, **c**, Nyquist plots and phase Bode plots, respectively, obtained from impedance spectroscopy for GelCA solutions and its composites with synthesized PEDOT additives at their dispersibility limits. Z_{img} and Z_{real} are the imaginary and real parts of the impedance. **d**, The results of fitting the impedance data to the equivalent circuit model are shown in the inset of **b**. R_i and R_p are resistances of their corresponding resistors in **b**, C_{dl} is the capacitive constant of capacitor C_{dl} , Q_p and n_p are the constant and exponential values of the capacitive phase element (CEP_p).

Comment 8. For Supplementary Figure 20, is the incubation time 24 hours as indicated in the scheme or 1 hour as indicated in the figure caption?

Response: We appreciate the Reviewer for pointing out this discrepancy. The incubation time was 24 hours, and we have corrected the figure caption in the revised manuscript.

Changes: The figure caption for the Supplementary Figure 20 was corrected as follows:

“Bioadhesive GelCA loaded with PEDOT:PSS and PEDOT:AlgS at their dispersibility limits after incubation for 24 h at 37 °C.”

Comment 9. Please remove the ‘e’ typo on Page 5, “... From a microstructural perspective, e scanning electron microscopy ...”.

Response: The ‘e’ typo was removed from the manuscript.

Changes (p. 6): The typo in the following sentence in section “Design and characterization of water-dispersible PEDOT conductive additives” was corrected:

“From a microstructural perspective, scanning electron microscopy (SEM) images (Figure 2d,e and Supplementary Figure 6) revealed that PEDOT:PSS aggregated excessively, leading to submillimeter-scale flake aggregates.”

Reviewer #2

General Comments: The manuscript presents a novel type of soft wearable electronics featuring PEDOT:AlgS with enhanced aqueous dispersibility and conductivity, integrated into various hydrogel systems. This study substantiates the design strategy that hydrophilic dopants can effectively modify the multiple properties of conducting polymers, leading to a broad spectrum of applications in biomonitoring. First, the conductive PEDOT-AlgS is hydrophilic, biocompatible, degradable, and can be easily prepared via 3D printing. Second, these additives are compatible with various hydrogel systems to construct wearable electronics with excellent physiological monitoring functions. These properties position PEDOT-AlgS as a promising candidate for implantable, biodegradable, and injectable bioelectronics. The manuscript is well-written, well-designed, and easy to follow.

Response: We thank the reviewer for very positive and constructive feedback for our manuscript. We have revised the manuscripts by addressing the suggestions noted by the reviewer as detailed below and believe that these changes have substantially improved our manuscript.

Comment 1. In the introduction section, the authors should explain why sulfonated alginate was chosen as the dopant for PEDOT.

Response: We sincerely appreciate the Reviewer’s positive feedback about our work. In the revised manuscript, we have clarified our rationale for using sulfonated alginate as a dopant for PEDOT. We propose that the hydrophobic nature of the PSS polystyrene backbone is responsible for the poor dispersibility of dry PEDOT. Therefore, alginate, which contains abundant polar groups, was selected as a hydrophilic backbone. To enable its doping capability, alginate was modified with sulfonate groups (sulfonated alginate, AlgS) to act as a doping agent, replacing PSS during the PEDOT polymerization process.

Changes (p. 3): The following statement was added to the last paragraph of the introduction:

“We hypothesize that hydrophobic polystyrene backbone of PSS contribute to the poor dispersibility of dry PEDOT:PSS. Thus, we chose alginate, with its rich content of polar groups, as a hydrophilic backbone and modified it with sulfonates (sulfonated alginate, AlgS) that serve as a doping agent in lieu of PSS in the PEDOT polymerization process.”

Comment 2. Hydrophilicity is one of the advantages of PEDOT:AlgS compared to conventional conductive polymers. Could the authors provide statistics on the water contact angles of both groups?

Response: We have performed contact angle experiments as per the Reviewer’s suggestion to further support the hydrophilicity of PEDOT:AlgS compared with PEDOT:PSS (see Figure R7) and provided statistics including standard deviation and p -values. These data suggest a statistically significant ($p < 0.05$) lower contact angle for PEDOT:AlgS as compared with PEDOT:PSS, which is attributed to enhanced hydrophilicity of dopant used during polymerization of PEDOT. We have added further discussion to highlight the results.

Figure R7. Water contact angle of PEDOT:PSS and PEDOT:AlgS coatings air-dried on glass substrates.

Changes: Supplementary Figure 7 was updated to reflect the contact angle measurements of Figure R7.

Supplementary Figure 7 | Characterization of PEDOT:AlgS re-dispersions in water after freeze-drying. **a**, Effects of sulfonation degree and EDOT monomer on the average hydrodynamic size results from DLS tests. **b**, Colloidal stability of PEDOT:AlgS measured in terms of the ratio of supernatant absorbance (after the 0.5% w/v solutions were left at rest for three months) to the absorbance of homogeneous solution of PEDOT:PSS and PEDOT:AlgS2 in water. **c**, Water contact angle of PEDOT:PSS and PEDOT:AlgS coatings air-dried on glass substrates. **d,e**, Viscosity-shear rate characteristics of PEDOT:AlgS and PEDOT:PSS solutions, respectively, in water at different concentrations. **f**, Comparison of viscosity-shear rate curves of commercial PEDOT:PSS and synthesized PEDOT:AlgS prepared at a similar PEDOT to dopant ratio (1:2.5) and dispersed at the same concentration in water (1.3 wt.%).

Further discussions were added to the “Design and characterization of water-dispersible PEDOT conductive additives” section to discuss the contact angle measurements (**p. 6**):

“This hydrophilicity was reflected in the contact angle results of Supplementary Figure 7c where a substantially lower contact angle was obtained in PEDOT:AlgS (26°) compared to PEDOT:PSS (48°).”

The Methods section was updated as follows (**p. 16**):

*“**Contact angle measurements.** Contact angle measurements were performed to assess the wettability of the surfaces using a goniometer RemaHart. A volume of 100 μ l PEDOT solution in water (2.5 % w/v) was dried on glass slides on hotplates at 60°C. A droplet of deionized water (~5 μ l) was carefully placed on the surface of coated layer using a microsyringe. Images of the droplet were captured upon contacting surface using a high-resolution camera, and the contact angle was determined by analyzing the droplet profile using ImageJ software.”*

Comment 3. Biocompatibility and biodegradability are two crucial challenges in implantable bioelectronics. The authors attributed the degradability of PEDOT:AlgS to the hydrolysis of glycosidic bonds in alginate. Is PEDOT:AlgS degradable in an in-vitro and cell-free environment, such as PBS or a cell culture medium at 37 °C? What is the complete degradation time both in vivo and in vitro? What are the primary degradation products and their potential metabolic pathways? Please provide more detailed information.

Response: In cell-free environments, such as PBS or cell culture media, PEDOT:AlgS is generally stable because it is not subject to enzymatic degradation by the enzymes unless alginate lyse or other degrading enzymes are introduced in the medium to catalyze the degradation process. We have performed in vitro molecular weight stability tests in PBS specifically to provide further evidence for the reviewer (Figure R8). The results indicate no visible peak shift to the right, indicating no substantial molecular weight reduction and thus no significant breakdown.

Figure R8. Results of SEC for PEDOT:AlgS before and after treatment in PBS for 11 weeks (PEDOT samples synthesized at 24 h EDOT polymerization).

To investigate in vitro degradability, we conducted an accelerated hydrolytic degradability test following the protocols reported previously^{R6} due to the hydrolysis rate being generally slow in neutral medium (see Figure 3i). The results of these studies suggest that, as opposed to PSS-doped PEDOT, molecular cleavage in AlgS group can occur due to hydrolysis of glycosidic bond. Although, determination of full molecular degradation time in vitro requires a much longer study and is an encouraged direction for future work.

Regarding in vivo degradation, as per the Reviewer’s suggestion we have now tested implantation of the PEDOT solutions over a longer period of 11 weeks (compared to one week in the previous version). As shown in Figure R9, the new results support progressive degradation and cell infiltration in PEDOT:AlgS while PEDOT:PSS structure remained stable and encapsulated in fibrotic tissues. Similar to the in vitro findings, complete degradation of PEDOT:AlgS in vivo would require longer implantation studies and we encourage these long-term timescales for future explorations.

Figure R9. Long-term in vivo degradation of intradermally injected PEDOT solutions. **a**, Images of injected PEDOT at different timepoints. **b**, H&E staining images of PEDOT:PSS and PEDOT:AlgS.

We note the PEDOT:AlgS is introduced as an additive to the hydrogel systems, and degradation in these systems are eventually controlled via hydrogel matrix parameters such as polymer concentration, crosslinking density, etc., depending on the intended application. However, these data suggest that regardless of the hydrogel matrix, removal of AlgS-doped PEDOT is substantially faster than PEDOT:PSS groups further supporting the previous claims of the paper. We have updated the manuscript with more discussions and additional data regarding degradation behavior of the polymers for further clarification.

Regarding the primary byproducts of degradation, although PEDOT is known to be generally stable, the small size distribution of PEDOT in PEDOT:AlgS due to the mitigated aggregations contributes to its enhanced removal from the body. Sulfonated alginate backbone can break into sulfonated oligo- and monosaccharides involving sulfonated mannuronic acid (M) and guluronic acid (G) residues according to hydrolysis-driven degradation reactions. Metabolic pathways of these small, sulfonated oligosaccharides are expected to be primarily through renal excretion. In this process, the sulfonate groups ($-\text{SO}_3\text{H}$) on the oligosaccharides remain intact and do not undergo substantial metabolic transformation in vivo. However, we demonstrated that these groups increase the water solubility of alginate and thereby enhanced solubility of its byproducts, which can facilitate renal clearance. Human enzymes are not known to metabolize sulfonated oligosaccharides, however some gut bacteria produce alginate lyases,^{R7} which can contribute further to the alginate degradation in vivo. We highlighted these potential byproducts and their metabolic pathways in the revised manuscript.

Changes: Supplementary Figure 9 was updated to include long-term in vivo implantation studies (see panels a and b):

Supplementary Figure 9 | Long-term in vivo degradation and immune response to intradermally injected 5 w/v% PEDOT solutions. a, Images of injected PEDOT at different timepoints. **b,** H&E staining images of PEDOT:PSS and PEDOT:AlgS. **c,** Immunostaining results using DAPI (blue), F4/80 (green), CD3, Ly6G, and CD80 (red) markers for week 3 after implantation. **d,** The ratio of F4/80⁺ cells to the total cells under the muscle layer (n=6).

Changes (p. 7 and p. 12): The following discussions were added to section “Biocompatibility and biodegradability of PEDOT solutions”:

“Longer implantation over 11 weeks showed progressive degradation in PEDOT:AlgS with cell infiltration while PEDOT:PSS remained stable within the fibrotic capsule (Supplementary Figure 9a,b).”

“To understand the degradation mechanisms, hydrolysis driven molecular weight changes of the polymers were tested in vitro (Figure 3i) suggesting that, unlike PEDOT:PSS, PEDOT:AlgS is molecularly degradable. The enhanced molecular degradability of PEDOT:AlgS is attributed to the hydrolyzable glycosidic bonds on the backbone of AlgS, which are absent in the PSS structure

(see Supplementary Figure 10). While PEDOT is generally stable, the byproducts of alginate are primarily alginate backbone broken into oligo- and monosaccharides involving sulfonated mannuronic acid (M) and guluronic acid (G) residues. The PEDOT phase is expected to remain intact due to the stable bonds, however, its smaller size distribution can promote their renal clearance in vivo. While sulfonation increases the hydrolyzability of alginate, as confirmed by Supplementary Figure 10b (aligning with previous studies⁴⁴), the enhanced solubility of alginate due to sulfonation can further facilitate its removal from the body. Metabolic pathways of these degradation byproducts are expected to be primarily through renal excretion. We note that although human enzymes do not metabolize sulfonated oligosaccharides, certain bacteria in the gut produce alginate lyases,⁴⁵ which may further contribute to alginate degradation through enzymatic cleavage of the glycosidic bonds.”

Changes (p. 13): The sentence below was added to the “Conclusions” section:

“Lastly, long-term in vivo studies of immune response and degradation, along with demonstrations of PEDOT-based hydrogels with active wound healing and antimicrobial properties are critical to expand their application for wound monitoring.”

Comment 4. In Figure 3, the Hematoxylin and eosin staining of intradermally injected solutions of PEDOT:AlgS and PEDOT:PSS showed that the PEDOT:AlgS group exhibits a loss of follicles and accumulation of fat tissues. Is this phenomenon due to the different implantation sites or the consequence of the different polymer components?

Response: We did not observe any consistent loss of follicles and fat accumulation in the PEDOT:AlgS groups, thus we attribute it solely to the different implantation site and not the material components. In addition to the new data presented on the longer-term implantation of the polymers (Figure R9), for the Reviewer’s reference, we have provided more H&E staining pictures of other repeats of week one from the PEDOT:AlgS and PEDOT:PSS conditions (Figure R10). Figure R10a shows an example of PEDOT:AlgS groups where the hair follicles are as dense as what shown in Figure 3h for PEDOT:PSS groups while thinner fat tissues can be seen close to the implantation site as opposed to the image shown for PEDOT:AlgS in Figure 3h. On the other hand, we also have images of PEDOT:PSS groups (Figure R10b) with fewer hair follicles in the epidermis layer as compared with PEDOT:PSS shown in Figure 3h. We have added more statements for clarification in the revised manuscript.

a

b

Figure R10. More images for the H&E staining results of **a**, PEDOT:AlgS and **b**, PEDOT:PSS.

Changes (p. 7): The following sentence was added to section “Biocompatibility and biodegradability of PEDOT solutions”:

“No meaningful differences in follicles or accumulation of fatty tissue were seen between the two groups.”

Comment 5. In the results section, the authors stated that “In vivo injection of 5% w/v PEDOT:PSS solutions resulted in the formation of a fibrous capsule around PEDOT:PSS, whereas immune cells infiltrated PEDOT:AlgS samples, attempting to digest the PEDOT polymers (Figure 3h).” It would be beneficial to identify the specific types of immune cells involved in this process. Further immune staining or flow cytometry (FACS) analysis is recommended to provide a more detailed characterization of the immune response to PEDOT:AlgS. Sequencing analysis might be valuable.

Response: As mentioned by the Reviewer, substantial cell infiltration was evident for the PEDOT:AlgS groups as opposed to PEDOT:PSS, which resulted in fibrous capsule formation. As we had already presented FACS analysis of immune response in vitro (Supplementary Figure 16), in the revised manuscript, we performed further immunostaining of additional in vivo experiments along with quantifications to determine the types of immune cells involved in this process with more details (Figure R11). The new findings indicate the presence of macrophages, neutrophils, and T cells among the infiltrated cells within the injected PEDOT:AlgS while no cell infiltration was observed for PEDOT:PSS and mostly macrophages were seen in the fibrotic capsule around it. More discussions are added to the manuscript to clarify these observations.

Figure R11. In vivo immune response to intradermally injected PEDOT solutions. **a**, Immunostaining results using DAPI (blue), F4/80 (green), CD3, Ly6G, and CD80 (red) markers for week 3 after implantation. **b**, The ratio of F4/80⁺ cells to the total cells under the muscle layer (n=6).

Changes: Supplementary Figure 9 was updated to include the immunostaining results of Figure R11 (see panels c and d).

Changes (p. 7): The discussions below were added to section “Biocompatibility and biodegradability of PEDOT solutions”:

“Immunostaining results suggest the limited presence of macrophages (F4/80⁺), neutrophils (Ly6G⁺), and T cells (CD3⁺) involved among the infiltrated cells in PEDOT:AlgS (Supplementary Figure 9c). Although, the number of immune cells constituted a substantially lower ratio of the present cells in PEDOT:AlgS compared to the PEDOT:PSS, which implies a strong immune response in PEDOT:PSS (Supplementary Figure 9d).”

Supplementary Figure 9 | Long-term in vivo degradation and immune response to intradermally injected 5 w/v% PEDOT solutions. a, Images of injected PEDOT at different timepoints. **b,** H&E staining images of PEDOT:PSS and PEDOT:AlgS. **c,** Immunostaining results using DAPI (blue), F4/80 (green), CD3, Ly6G, and CD80 (red) markers for week 3 after implantation. **d,** The ratio of F4/80⁺ cells to the total cells under the muscle layer (n=6).

The following details were added to Methods section under “In vivo immunofluorescence staining” (p. 21 and p. 22):

“Anti-mouse F4/80, CD80, CD3, and Ly6G, as well-established marker for identifying mouse macrophages, their pro-inflammatory subtypes, T cells, and neutrophils were used.”

“Green for F4/80 macrophages, red for CD80, CD3, and Ly6G”

Comment 6. In Supplementary Figure 9, the immunostaining of F4/80 is unclear. Please provide higher magnification images for better visualization. Additionally, including statistical analysis of the F4/80 staining results would enhance the clarity and impact of the findings.

Response: We agree that higher magnification images would improve the clarity and visualization of the F4/80 positive macrophages. To address this issue, we performed new experiments and imaged the slides at higher magnification (20x compared to 5x reported previously) to provide better resolution and detail in the revised manuscript, as shown in Figure R11a. The results reconstituted the previous claims with better data quality. In addition, we have quantified the F4/80 positive cells across multiple sections and presented the statistical analysis, including the mean values and standard deviations, along with appropriate statistical tests to evaluate significance (Figure R11b).

Changes: The data in Supplementary Figure 9 was replaced with updated high magnification images of Figure R11 for better clarity (see panels c and d):

Changes (p. 7): The discussions below were revised in section “*Biocompatibility and biodegradability of PEDOT solutions*” to reflect the changes explained above:

“Immunostaining results suggest the limited presence of macrophages (F4/80⁺), neutrophils (Ly6G⁺), and T cells (CD3⁺) involved among the infiltrated cells in PEDOT:AlgS (Supplementary Figure 9c). Although, the number of immune cells constituted a substantially lower ratio of the present cells in PEDOT:AlgS compared to the PEDOT:PSS, which implies a strong immune response in PEDOT:PSS (Supplementary Figure 9d).”

Supplementary Figure 9 | Long-term in vivo degradation and immune response to intradermally injected 5 w/v% PEDOT solutions. a, Images of injected PEDOT at different timepoints. **b,** H&E staining images of PEDOT:PSS and PEDOT:AlgS. **c,** Immunostaining results using DAPI (blue), F4/80 (green), CD3, Ly6G, and CD80 (red) markers for week 3 after implantation. **d,** The ratio of F4/80⁺ cells to the total cells under the muscle layer (n=6).”

Comment 7. The pH sensing property is a significant highlight of the GelCA-PEDOT-AlgS bioadhesive application. The authors attributed the pH sensing mechanism in GelCA-PEDOT to catechol-to-quinone transitions by CA. This assumption should be supported by appropriate compositional analysis. Moreover, as catechol is easily oxidized in air, the pH sensing accuracy and sustainability are questionable. The authors should test the pH sensing accuracy at different time points after implantation to indicate the validity period of the bioelectronic device.

Response: We are happy that the Reviewer brought up this important point. After a more in-depth investigation of the underlying reasons for the trends observed in current sensitivity to pH, we

found that in addition to redox reactions driven by catechol groups, pH-dependent reversible swelling of the hydrogel matrix (leading to enhanced ion mobility) could also be responsible for the observed proportional trends between pH and current. Previous papers reporting conductivity of pH-responsive non-catecholic hydrogels^{R8,R9} have shown similar trends even in the absence of PEDOT or other conductive additives. We have modified the schematic of Figure 5g in the updated form of Figure R12 and have updated our discussions in the main manuscript.

Figure R12. Mechanisms of pH-sensitive current changes in hydrogels including possibilities of redox reactions and hydrogel swelling leading to enhanced ion and electron mobility.

Since the catechol compositional analysis suggested by the Reviewer can influence hydrogel swelling in addition to redox-driven effects, the results of those studies would not allow for a clear decoupling of these effects. Our original argument on possible contribution of catechol-driven redox reactions was based on discussions of Ref R10, as referenced in the manuscript. Regarding accuracy and sustainability due to these reactions, we first note that according to Figure 5g, antibiotic treatment of infected wound recovered the current change ratio suggesting that the irreversible effects of potential catechol oxidation due to exposure to air, if there are any, are negligible. Note that catechol was conjugated to GelCA in its oxidized oligomerized form, which is more stabilized and in this form, redox quinone to catechol reverse reactions are more favored. Nevertheless, we performed more experiments to characterize durability of sensors over time as the Reviewer suggested. Here, the *in vivo* wound pH response to infection was assessed after three days past introducing the skin punctures to allow for possible permanent catechol oxidations (Figure R13). The results indicated similar trends suggesting the utility of the sensors in time scales relevant to wound healing process. In addition, we performed an *in vitro* signal sustainability study to eliminate variabilities due to presence on the wound (Figure R14). These results also did not show a statistically meaningful correlation between pH and ultimate conductivity variations over one week. The manuscript was updated to reflect the above findings.

Figure R13. Current stability of the crosslinked GelCA+PEDOT:AlgS hydrogels at different pH levels.

Figure R14. In vivo sensing of wound infection after 3 days post wound closure.

Changes: Figure 5g was updated to correct the envisioned pH sensing mechanism of Figure R12:

Figure 5 | Injectable conductive bioadhesives for implantable bioelectronics applications. a, Formulation of bioadhesives involving gelatin-catechol (GelCA), synthesized *via* coupling caffeic acid to gelatin, as hydrogel bioadhesive matrices. PEDOT doped with PSS and AlgS are incorporated separately within GelCA at their dispersibility limits (4 and 20% w/v, respectively) and crosslinked ionically by Fe^{3+} for pH sensing in wound monitoring applications. **b,** Hydrogel prepolymer composites of GelCA (12% w/v) with PEDOT:PSS (4% w/v) and PEDOT:AlgS (20% w/v) after shaking on a vortex for 1 min. **c,** Viscosity-shear rate of injectable bioadhesives. **d,** Impedance spectroscopy of injectable bioadhesive prepolymers. **(E)** *Ex vivo* porcine lung burst pressure adhesion testing of hydrogels. **f,** Clotting time assays for the assessment of hemostatic activity after hydrogel crosslinking. **g,** *In vitro* pH sensitivity of conductive bioadhesives obtained by chronoamperometric testing of hydrogels in various pH levels. The inset shows current variations with time at pH 7. **h,** *In vivo* monitoring of wound infection using conductive bioadhesive hydrogels.

Supplementary Figure 22 was updated to add the *in vitro* and *in vivo* sustainability data of Figure R13 and Figure R14 (panels d and e, respectively):

Supplementary Figure 22 | pH sensing capabilities of conductive bioadhesives. **a,b,** Real-time data of chronoamperometry tests showing the dependence of conductivity to different pH values for GelCA (12% w/v) + PEDOT:PSS (4% w/v) and + PEDOT:AlgS (20% w/v), respectively. The tests were conducted in buffer solutions with varying pH levels (pH 4, 7, and 9) while applying a potential of 0.65 V. **c,** Open circuit potential results comparing pH sensitivity of bioadhesives. **d,** Current stability of the crosslinked GelCA+PEDOT:AlgS hydrogels at different pH levels. **e,** In vivo sensing of wound infection after 3 days post wound closure. CE, counter electrode; WE, working electrode; RE, reference electrode.

The following discussions were added to section “Injectable smart bioadhesives for wound monitoring” (p. 12):

“The pH sensing mechanism in GelCA-PEDOT is attributed to synergistic effects of mediated electron and ion mobility due to hydrogel swelling with as well as catechol-to-quinone transitions with pH due to oxidation reactions⁵².”

“Stability of conductivity over one week in various pH levels suggested the negligible effects of non-reversible reactions such as catechol oxidation (Supplementary Figure 22d).”

“The current response to wound infection was also tested three days after introducing skin wounds where similar trends were observed indicating sensing sustainability in wound healing timescales (Supplementary Figure 22e).”

Comment 8. A major concern in the application of hydrogel-based wound monitors is the risk of infection, which can hinder effective treatment. Therefore, the hydrogel is expected to possess anti-infective and wound healing properties. It is crucial to evaluate whether PEDOT-AlgS, either alone or in a hydrogel system, can exert a therapeutic effect on infectious wound healing. Please examine the effect of PEDOT-AlgS on skin healing status and its anti-infective properties.

Response: We agree that infection is a major concern in hydrogel wound sensors. Hydrogels are in fact excellent platforms to incorporate antibacterial and wound-healing functionalities via additives or by functionalizing their polymer networks with the appropriate building blocks. These hydrogel design strategies have been heavily explored in the literature^{R11,12} and are not envisioned in the presented materials design. The proposed PEDOT:AlgS additives aim to address the limited water dispersibility of electroactive PEDOT:PSS additives as an additive for injectable bioelectronics. Nevertheless, we have performed separate antibacterial and wound-healing assays for the reviewer's reference (Figure R15 and Figure R16, respectively). Our results indicate no substantial difference in antiseptic and wound healing activity of PEDOT:PSS and PEDOT:AlgS. A negligible zone of inhibition is observed when the hydrogel prepolymers are crosslinked with FeCl₃, which is suggestive of antibacterial effects due to oxidative stress. We have clarified in the manuscript that stronger wound healing and antibacterials properties could be incorporated through additional functionalization of the hydrogels or by selecting specific hydrogel matrices. We believe these enhancements are promising directions for future research, and we encourage future exploration along these lines.

Figure R15. Wound-healing effects of PEDOT:PSS and PEDOT:AlgS additives. (a) Contraction of wound edges treated with electroconductive hydrogels. (b) Quantification of wound area after 6 and 12 days post wound healing.

Figure R16. Antibacterial zone of inhibition for PEDOT:PSS and PEDOT:AlgS additives in GelCA before and after crosslinking via FeCl_3 against *E. coli* bacterial strains.

Changes (p. 12): The following statement was added to section “Injectable smart bioadhesives for wound monitoring”:

“Likewise, PEDOT additives did not show substantial antibacterial effects nor affect wound healing properties in vivo (Supplementary Figure 24).”

The sentence below was also added to the Conclusions section (p. 13):

“Lastly, long-term in vivo studies of immune response and degradation, along with demonstrations of PEDOT-based hydrogels with active wound healing and antimicrobial properties are critical to expand their applications in wound monitoring.”

Also, Figure R15 and Figure R16 are combined in the form of Supplementary Figure S24 in the revised manuscript as shown below.

More sections were added to the Methods section (p. 23 and p. 24):

“Antibacterial tests. The antimicrobial activity of hydrogels was evaluated using an agar well diffusion assay. *E. coli* Microbial strains were cultured in nutrient broth until reaching an optical density (OD) at 600 nm of 1.24 for *E. coli* measured with a UV-Vis spectrometer. Agar plates were prepared by spreading the inoculum over the surface. Pre-gel solutions (100 μl) were pipetted on the agar plates. Crosslinked hydrogels were placed directly on the agar surface. The plates were incubated at 37 $^\circ\text{C}$ for 24 h to observe the zones of inhibition. As a positive control, 8 mm filter paper disks impregnated with 15 μg of silver sulfadiazine were placed on the agar plates.

In vivo wound healing. Circular full-thickness wounds were created on the dorsal skin of the animal model using an 8 mm biopsy punch. After wound creation, the wounds were treated with the designated hydrogels and covered as needed to maintain the treatment. Wound area measurements were taken on days 6 and 12 post-treatment using images captured to track the healing process.”

Supplementary Figure 24 | Wound healing and antibacterial effects of PEDOT additives in bioadhesive GelCA hydrogels. a, *In vivo* healing of skin wounds after 6 and 12 days using electroconductive bioadhesive dressings. **b**, Relative decrease in wound area calculated based on the wound traces. **c**, Zone of inhibition antibacterial tests against *E. Coli* bacterial strain for GelCA bioadhesives supplemented with PEDOT:PSS and PEDOT:AlgS before and after crosslinking.

Comment 9. In the Methods section, please specify the sample size, sex, age, and ethical approval of the animals used in the study.

Response: We double-checked the Methods section and included all the missing specifications related to animals used.

Changes (p. 21 and p. 23): More details were given in Methods section under “*In vivo* biocompatibility” and “*In vivo* pH sensing tests” in the following paragraphs, respectively:

“...following an approved protocol at the Lundquist Institute (#22747-01). For each sample, five black male C57BL/6 mice, 6-8 weeks old, obtained from The Jackson Laboratory in the USA, were housed in standard laboratory conditions.”

“The animal experiment was performed using male, 9-12 weeks old, ob/ob mice, B6.Cg-Lepob/J mice (The Jackson Laboratory, Bar Harbor, ME, USA) according to the protocol approved by the Institutional Animal Care and Use Committee (protocol no. IA23-1800) at California Institute of Technology.”

Comment 10. Please consider expanding the discussion on the potential future applications of injectable conductive hydrogel monitoring devices. Including a dedicated section in the conclusion or discussion that outlines the prospects, challenges, and possible advancements in this field would provide valuable insights for readers and highlight the broader impact of your work.

Response: Thank you for your thoughtful suggestion. We agree that a more detailed discussion on the future applications of injectable conductive hydrogel monitoring devices would enhance the manuscript. In response, we have expanded the discussion section to include potential future applications, such as real-time health monitoring, soft bioelectronics, and neural interfaces. We also address key challenges, such as long-term biocompatibility, integration with existing medical technologies, and scalable manufacturing methods. Additionally, we have highlighted possible advancements in materials design and functionality that could improve the performance and adaptability of these devices. These additions emphasize the broader impact of our work and provide insights into the future directions of this promising field.

Changes (p. 13): The following paragraph was added to the Conclusions section:

“Overall, doping conducting polymers with hydrophilic moieties such as AlgS holds promising potential for applications in injectable bioelectronics. We envision that various natural biomolecules can be modified for doping polymeric semiconductors to design bioactive electrical conductors that enables more efficient interfacing with tissues and engineering biomaterials such as soft hydrogels. Integrability of these additives with other crosslinking systems such as free-radical photopolymerization further expands their utility in development of minimally invasive theranostics. Besides, despite advances in improving conductivity, hydrogel-based electrodes still exhibit substantially lower conductivity compared to metallic electrodes, highlighting the need for further innovations in this space. Continued efforts in this field will open new avenues for sophisticated injectable bioelectronic systems capable of long-term in vivo monitoring and therapeutic interventions for more applications in real-time health monitoring and neural interfacing.”

Reviewer #3:

General Comments. The manuscript authored by Wei Gao and co-workers reports the synthesis of conductive hydrogels made of PEDOT doped with sulfonated alginate (AlgS) to be used as functional electrodes for soft injectable electronics. The idea is interesting, but there are several errors (Error! Reference source not found) that make it impossible to read the manuscript and relate the text with the figures for a correct evaluation of the work. Therefore, the authors need to solve these issues and resubmit the manuscript so that it can be reviewed and evaluated.

Response: We thank the Reviewer for the positive words and sincerely apologize for the technical issues (e.g., “Error! Reference source not found”) that occurred in the manuscript, which disrupted the review process. These errors were likely introduced during the file conversion or submission

process, resulting in broken references to figures and sections. We have thoroughly reviewed the manuscript and corrected all the reference errors to ensure that the figures, sections, and citations are properly linked. The revised version of the manuscript has been resubmitted, and we hope that it is now fully readable and can be evaluated without any issues.

Changes: The broken links marked by “Error! Reference source not found” in the entire text were corrected throughout the text.

References

- R1. Hsieh J-C, He W, Venkatraghavan D, Koptelova VB, Ahmad ZJ, Pyatnitskiy I, *et al.* Design of an injectable, self-adhesive, and highly stable hydrogel electrode for sleep recording. *Device* **2**, 100182 (2024).
- R2. Jin S, Choi H, Seong D, You C-L, Kang J-S, Rho S, *et al.* Injectable tissue prosthesis for instantaneous closed-loop rehabilitation. *Nature* **623**, 58-65 (2023).
- R3. Kim S, Jang J, Kang K, Jin S, Choi H, Son D, *et al.* Injection-on-Skin Granular Adhesive for Interactive Human–Machine Interface. *Adv. Mater.* **35**, 2307070 (2023).
- R4. Park J, Lee S, Lee M, Kim H-S, Lee JY. Injectable Conductive Hydrogels with Tunable Degradability as Novel Implantable Bioelectrodes. *Small* **19**, 2300250 (2023).
- R5. Sakunpongpitiporn P, Phasuksom K, Paradee N, Sirivat A. Facile synthesis of highly conductive PEDOT:PSS *via* surfactant templates. *RSC Adv.* **9**, 6363-6378 (2019).
- R6. Volpatti LR, Bochenek MA, Facklam AL, Burns DM, MacIsaac C, Morgart A, *et al.* Partially oxidized alginate as a biodegradable carrier for glucose-responsive insulin delivery and islet cell replacement therapy. *Adv. Healthcare Mater.* **12**, 2201822 (2023).
- R7. Rønne Mette E, Madsen M, Tandrup T, Wilkens C, Svensson B. Gut bacterial alginate degrading enzymes. *Essays in Biochemistry* **67**, 387-398 (2023).
- R8. Wang T, Zhang X, Wang Z, Zhu X, Liu J, Min X, *et al.* Smart Composite Hydrogels with pH-Responsiveness and Electrical Conductivity for Flexible Sensors and Logic Gates. *Polymers* **11**, 1564 (2019).
- R9. Sheppard NF, Lesho MJ, Tucker RC, Salehi-Had S. Electrical conductivity of pH-responsive hydrogels. *J. Biomater. Sci., Polym. Ed.* **8**, 349-362 (1997).
- R10. Odinotski S, Dhingra K, GhavamiNejad A, Zheng H, GhavamiNejad P, Gaouda H, *et al.* A conductive hydrogel-based microneedle platform for real-time pH measurement in live animals. *Small* **18**, e2200201 (2022).
- R11. Jia B, Li G, Cao E, Luo J, Zhao X, Huang H. Recent progress of antibacterial hydrogels in wound dressings. *Mater. Today Bio* **19**, 100582 (2023).
- R12. Gounden V, Singh M. Hydrogels and Wound Healing: Current and Future Prospects. *Gels* **10**, 43 (2024).

Responses to the reviewers' comments

We sincerely appreciate the constructive feedback from all three reviewers. In response, we have made further revisions to address the remaining comments from Reviewer 3. The revised text is highlighted in yellow in the main text and below is a response to each comment.

Reviewer #1:

The authors have revised their manuscript to address clearly the issues raised by the reviewer. The reviewer now recommends the manuscript for publication in this journal.

Response: We sincerely appreciate the Reviewer's positive feedback on the revised manuscript and their recommendation for publication.

Reviewer #2:

The authors have effectively addressed the concerns raised by providing solid evidence. Therefore, I recommend the acceptance of this manuscript.

Response: We sincerely appreciate the Reviewer's positive feedback on the revised manuscript and their recommendation for publication.

Reviewer #3:

General Comment. This manuscript reports the synthesis of hydrophilic PEDOT dispersions by doping this conducting polymer with sulfonated alginate (AlgS) instead of PSS. This increased the conductivity of the PEDOT:AlgS dispersions compared to commercial PEDOT:PSS. The authors carried out an in-depth study of the properties of this dispersion, as well as of its use in electrically functionalized adhesive gelatin hydrogels for biosensing purposes. Although the idea presented here is interesting and the work is well conducted, the following points need to be clarified to improve the quality of the manuscript.

Response: We sincerely appreciate the Reviewer for re-evaluating our paper, providing positive feedback, and offering constructive suggestions. In response, we have revised the manuscript to enhance the presentation and clarity of our work, incorporating the suggested improvements as outlined below.

Comment 1. On page 5 it is mentioned that "While sulfonation interrupted alginate ionic associations, possibly due to steric hindrance and conformation changes, comparable mechanical properties could be achieved by increasing AlgS content, leveraging its enhanced solubility." Therefore, why was the elastic modulus of AlgS2 hydrogels lower than that of AlgS1 hydrogels?

Response: The storage modulus results shown in Supplementary Figure 2c,d were obtained by testing sulfonated alginates at their solubility limit, which was determined qualitatively based on selected concentration increments (Supplementary Figure 2b).

Due to the high solubility of AlgS2, even at 30% w/v (marked as partially soluble), the AlgS2 solutions remained less viscous (as observed in the vial tilt experiments) than the partially dissolved 15% w/v AlgS1. Therefore, while 20% w/v was reported as the solubility limit for AlgS2, this value is approximate, as the next tested concentration increment was 30% w/v. Further concentrating AlgS2 beyond 20% w/v would likely yield a storage modulus closer to that of AlgS1.

To test this idea and to support this statement, we have now measured the ionic crosslinking kinetics for AlgS2 at 25% w/v, and the results further reinforce this observation (see Figure R1). However, to maintain consistency in solubility limit testing across different sulfonation degrees, these results are provided for the Reviewer’s reference only. To enhance clarity and to avoid potential confusion, we have revised the manuscript as noted below.

Figure R1. Rheological characterization of ionic crosslinking kinetics for AlgS2 at 20% and 25% w/v in response to 100 mM FeCl₃.

Changes (p. 5): The following statement was revised in section “Design and characterization of water-dispersible PEDOT conductive additives”:

“While sulfonation interrupted alginate ionic associations, possibly due to steric hindrance and conformational changes, comparable mechanical properties can still be achieved by increasing the AlgS content, leveraging its enhanced solubility.”

Comment 2. I suggest calling “sulfonated alginate-doped PEDOT polymers” instead of “PEDOT-doped sulfonated alginate polymers”

Response: Thank you. We have replaced the term “PEDOT-doped sulfonated alginate” with “sulfonated alginate-doped PEDOT” in the manuscript.

Changes (p. 34): Figure caption title in Figure 2 was revised as follows:

“Characterization of freeze-dried sulfonated alginate-doped PEDOT polymers.”

Comment 3. I suggest showing the whole wavelength range (225 – 1200 nm) for all UV-vis spectra to establish a clear comparison of the different conditions tested in the same range (Figure 2a and Supplementary Figure 3a-c).

Response: The wavelength ranges in these figures have been selected to highlight the regions of interest from the full spectra that is relevant to our specific discussions of either polymerization kinetics or doping states in PEDOT polymers. We tried full spectra representation of UV-vis data; however, the plot scales to contain full range compacts the peaks of interest, making it difficult to highlight comparisons between conditions. Therefore, we have kept the low and high wavelength ranges separately for better communication with the reader and for more clarity.

Comment 4. What was the conductivity of PSS-doped PEDOT after 2 d (Supplementary Figure 4)?

Response: PEDOT:PSS is our control condition in Supplementary Figure 4, which is typically synthesized through one-day polymerization of EDOT monomers in the standard protocols.^{R1,R2} Given the different polymerization rates of PEDOT when doped with PSS versus AlgS, the purpose of this experiment was to determine the optimal polymerization time for PEDOT:AlgS such that its conductivity matches that of standard PEDOT:PSS, enabling correct comparisons in subsequent characterizations such as dispersibility. Thus, we initially did not consider additional time points for the synthesis of PEDOT:PSS. However, to address the Reviewer's suggestion, we performed PEDOT:PSS synthesis following our protocol and extended the reaction to two days. This protocol resulted in a viscous solution with inhomogeneous aggregates, which produced cracked films upon drying the solution on glass slides (Figure R2 for Reviewer's reference).

Figure R2. Drying process of PEDOT:PSS after a 2-day polymerization reaction. a, Appearance of the reaction solution. **b,** Formation of a cracked film formation after the drying process.

These structural defects prevented reliable conductivity measurements, as the films lacked sufficient integrity for accurate characterization. The manuscript is revised for further clarification in this regard.

Changes (p. 5): The following statements were added to section “Design and characterization of water-dispersible PEDOT conductive additives”:

“Given the differences in polymerization kinetics between PSS and AlgS dopants, the polymerization time for PEDOT:AlgS2 was set to two days to achieve a dry conductivity comparable to standard PEDOT:PSS controls, which are typically synthesized over one day.”^{38,41}”

Comment 5. How were the authors able to distinguish between PEDOT and AlgS in the SEM images (Figure 2e and Supplementary Figure 6)? A chemical scanning technique must be used to

distinguish between two different chemical components in the topography (for example, nanoFTIR).

Response: It is well established that hydrophilic polymers such as AlgS and PSS form 3D porous morphologies in aqueous media, whereas hydrophobic moieties like PEDOT tend to form aggregates. In our SEM images, which include either PEDOT:PSS or PEDOT:AlgS systems, we observed clear aggregates dispersed on porous structures (see low magnification images in Figure 2e). These morphological differences enabled us to attribute the phases based on their structures. Furthermore, the observed aggregate sizes in SEM align with our DLS results, supporting the structure-morphology correlation.

In response to the Reviewer's suggestion, we explored additional approaches to characterize phase distribution in the SEM images. While nano-FTIR, as suggested, combines AFM with infrared spectroscopy, it is primarily suitable for surface coatings or fixed surfaces. Our samples are freeze-dried PEDOT in the form of loose powders, making nano-FTIR impractical. We also considered EDX and Raman microscopy. However, EDX is not reliable in this case, as the chemical elements in PEDOT, PSS, and AlgS are common across all components, precluding phase identification. Similarly, Raman microscopy, with a resolution slightly below the micron level, is insufficient for resolving PEDOT aggregates, which are ~100 nm in size in the SEM images.

Due to these limitations, we have revised the discussion to acknowledge the constraints of our analysis explicitly. We also encourage future studies to employ advanced chemical characterization methods, such as Time-of-Flight Secondary Ion Mass Spectrometry (ToF-SIMS), which was not available at our accessible facilities.

Changes (p. 6): The following sentences were revised in section "Design and characterization of water-dispersible PEDOT conductive additives":

"In these images, the observed phases (PEDOT, PSS, and AlgS) were identified solely based on their distinct morphologies, with porous structures corresponding to hydrophilic polymer phases (dopant) and aggregates corresponding to PEDOT. While these results align well with dynamic light scattering (DLS) data (Figure 3a), further chemical analyses are required to validate the identified phase attributions."

Comment 6. The molecular weight of the sodium alginate used needs to be determined, as it may explain the differences observed with PSS (70,000 Da).

Response: The molecular weight of alginate used in this study has been determined elsewhere^{R3} by gel permeation chromatography (GPC) and reported to be 187 kDa. Since the molecular weight of alginate is higher than PSS (70 kDa) and as our results show no significant difference in GPC data among alginate and their sulfonated compounds (Supplementary Figure 2a), we could not attribute the enhanced dispersibility of PEDOT:AlgS to the molecular weight of dopants. We have added more details on the alginate used in this study for clarification.

Changes (p. 14): The catalog number for sodium alginate used in this study is provided in “Materials and reagents” section:

“sodium alginate (cat. no. W201502)”

Changes (p. 4): The alginate used in this study and its molecular weight was highlighted in section “Design and characterization of water-dispersible PEDOT conductive additives”:

“alginate (W201502, ~200 kDa³⁶)”

Comment 7. What happens to the mean hydrodynamic diameter of PEDOT:PSS at different concentrations after freeze-drying (Supplementary Figure 7a)?

Response: Supplementary Figure 7a represents the effects of alginate sulfonation on the average hydrodynamic diameter in PEDOT:AlgS. The mean hydrodynamic diameter for the PEDOT:PSS controls along with their size distribution is presented in Figure 3b and is compared with corresponding synthesis conditions of PEDOT:AlgS. Note that as a control group, we considered PEDOT:PSS synthesized only at the maximum EDOT concentration (*i.e.*, 0.9% w/v at which PEDOT:AlgS was also synthesized) throughout the paper for its maximized conductivity among all conditions. Hence, various EDOT concentrations were not relevant in the context of this paper and such effects have been extensively explored in the literature.^{R1}

Comment 8. What is the meaning of the inverted eppendorf shown in Figure 3e?

Response: The inverted tubes shown in Figure 3e show that freeze-dried PEDOT:PSS in water at 10% w/v self-associates while a much higher concentration (20% w/v) of freeze-dried PEDOT:AlgS in water forms a dispersion with low enough viscosity that can easily flow under gravity. This result compares loading capacity of PEDOT in water when doped with PSS versus AlgS, which is indicative of better injectability of PEDOT:AlgS compared with PEDOT:PSS in aqueous systems. We have made revisions in the revised manuscript for further clarification.

Changes (p. 35): The following clarifications were made in figure caption 3e:

“Tubes represent 10% w/v PEDOT:PSS in water forming self-associated gels and 20% w/v PEDOT:AlgS remaining in liquid phase.”

Changes (p. 7): The following statement in section “Aqueous re-dispersion of PEDOT-based conductive polymers” was revised as follows:

“The dispersibility limits of PEDOT:AlgS was improved with sulfonation of alginate (Figure 3e) exceeding those of PEDOT:PSS by about 4–5×, as the hydrophilic backbone of alginate facilitated interactions with water molecules.”

Comment 9. To prove that PEDOT:AlgS is ionically responsive to Fe³⁺ (Figure 3g), the evolution of the elastic modulus of the PEDOT:AlgS2 dispersion over time without contact with FeCl₃ should be shown.

Response: We appreciate this suggestion. We have tested PEDOT:AlgS rheological data in the absence of Fe³⁺ showing no gelation thereby more strongly supporting the ionic responsiveness of

the presented materials (Figure R3). Those data will not change the discussions and conclusions of the paper and are now added to the revised manuscript.

Figure R3. Rheological variations of PEDOT:AlgS2 in the absence of Fe^{3+} ions.

Changes: The data of Figure R3 are added to Supplementary Figure 8a:

Supplementary Figure 8 | Ionic responses of PEDOT:AlgS synthesized at various EDOT concentrations to Fe^{3+} . **a**, Rheological variations of PEDOT:AlgS2 in the absence of Fe^{3+} ions. **b**, Results represent response to Fe^{3+} diffusion in terms of storage modulus (G') after 5 min post-treatment with 100 mM FeCl_3 solutions. The PEDOT:AlgS solutions were prepared at their dispersibility concentration limit. The data represent mean \pm standard deviation ($n=3$).

Comment 10. How long did the hydrolytic degradation take to obtain the results shown in Figure 3i and Supplementary Figure 10? The differences between the spectra before and after hydrolysis are not large.

Response: Given the long degradation times for hydrolysis of alginate backbone, hydrolytic degradation was conducted for three weeks in accelerated conditions following previous protocols.^{R4} We have added more details in methods section to clarify the in vitro biodegradation testing procedure.

For the PEDOT:PSS and PSS samples, no differences before and after hydrolysis were expected due to the stable bonds. Larger molecular weight species are associated with lower retention times in these plots and as shown in Figure 3i, for PEDOT:AlgS2, there is a prominent drop in the first peak (at low retention time of ~3.5 min), which is absent in PEDOT:PSS. The peak shifts in the alginate and its derivatives (Supplementary Figure 10b), although less strong, they are larger than PSS controls. We note that data in Supplementary Figure 10b is provided to understand contribution of dopant in overall degradation seen in Figure 3i and given that molecular weights and retention times are logarithmically correlated, we conclude a meaningful difference in degradability of alginate-based products. We made the following revisions in the manuscript to highlight the above discussions.

Changes (p. 17): The following sentences were added to section “Size-exclusion chromatography”:

“For accelerated hydrolytic degradability tests, the products were treated in 20 mM NaOH at 5% w/v concentration and maintained at 37 °C for three weeks. Before the SEC tests, the samples were diluted to 0.5 mg ml⁻¹ using DI water.”

Changes (p. 8): The following sentence was added:

“To understand the degradation mechanisms, hydrolysis driven molecular weight changes of the polymers were tested in vitro (Figure 3i). The decay seen in the peak of PEDOT:AlgS, which is absent in PEDOT:PSS, suggests that, unlike PEDOT:PSS, PEDOT:AlgS is hydrolytically degradable.”

Comment 11. On page 8, it is mentioned that coatings were formed with the 2.5% w/v solutions) (Figure 3jII and Supplementary Figure 11c–e), but in the caption of Supplementary Figure 11c–e it is written that AFM imaging of coatings obtained from 5% w/v PEDOT solutions. The same concentration should be used in all cases to establish a comparison between samples.

Response: We greatly appreciate the Reviewer for bringing this discrepancy to our attention. After carefully looking at our notes and raw data, we confirm that the concentration mentioned in the manuscript (2.5% w/v) was correct and the figure captions (both Figure 3j and Supplementary Figure 11c–e) have now been corrected accordingly in the revised manuscript.

Changes (p. 35): Figure caption of Figure 3jII was corrected as follows:

“glass substrate and their II: atomic force microscopy (AFM) phase plots obtained from dispersions (2.5% w/v).”

Changes: Figure caption of Supplementary Figure 11d,e was corrected as follows:

“The z-plots and phase plots, respectively, for PEDOT:AlgS2 and PEDOT:PSS samples from AFM imaging of coatings obtained from 2.5% w/v PEDOT solutions.”

Comment 12. What about the comparison of the conductivity properties of the current work (Supplementary Figure 12) with those of reference 25 of the main draft (Yuk H, Lu B, Lin S, Qu K, Xu J, Luo J, et al. 3D printing of conducting polymers. Nat. Commun. 11, 1604 (2020)?

Response: We appreciate the reviewer bringing up this important reference, which is a seminal work on 3D printing of pure PEDOT:PSS hydrogels. Our study focuses on injectable hybrid hydrogels broadly where the use of PEDOT is required as an additive to enhance the electrical conductivity of existing hydrogel systems. Hence, we structured Supplementary Figure 12 in terms of the relative increase in hydrogel matrix conductivity where we normalize for matrix effects to better isolate and compare the contribution of PEDOT in enhancing hydrogel conductivity. The referenced work examines pure PEDOT:PSS hydrogels, where the concept of relative conductivity increase within a matrix does not apply. However, given the significance of this reference, we have expanded the discussion in the revised manuscript to contextualize our findings in relation to their outcomes.

Changes (p. 9): The following discussions were added to section “Hydrogel-based 3D printed soft bioelectrodes”:

“Given the electrically insulating nature of existing hydrogels (e.g., alginate with a conductivity of $\sim 7.1 \times 10^{-4} \text{ S m}^{-1}$), the conductivity of PEDOT:AlgS-incorporated hydrogels ($\sim 7.5 \times 10^{-2} \text{ S/m}$) is lower than reports on pure PEDOT hydrogels (on the order of 10^{-3} – 10^{-5} S m^{-1} ^{22,25}). However, it is important to note that processing pure PEDOT hydrogels typically requires drying steps and the use of organic solvents or cytotoxic ions, which restricts their applicability in scenarios where direct injectability is required.”

Changes (p. 36): We added matrix conductivity used to normalize conductivities in the figure captions of Figure 4c and Supplementary Figure 19a, respectively:

“The relative increase in conductivity of alginate ($7.1 \times 10^{-4} \text{ S m}^{-1}$) as a result of introducing conductive PEDOT fillers”

“The maximum relative increase in conductivity of 12% w/v GelCA matrices ($2.9 \times 10^{-4} \text{ S m}^{-1}$) attainable with PEDOT:AlgS and PEDOT:PSS.”

Comment 13. On page 9, it is mentioned that in terms of printing fidelity (Figure 4e), solutions of 4% w/v PEDOT:PSS in alginate experienced multiple clogging events due to aggregation and precipitation, but these results are not shown in Figure 4e. I suggest including the results concerning to 3D printing tests of PEDOT:PSS + alginate in the supporting information file for comparison with those of PEDOT:AlgS + alginate shown in Figure 4e.

Response: As per the Reviewer’s suggestion, we performed additional experiments to include more images of printing fidelity and to support our observations of clogging events in the case of PEDOT:PSS. As shown in Figure R4, we loaded the inks into syringes while introducing some air between the ink columns. As the ink is pushed out by air pressure, the alginate+PEDOT:AlgS is continuously dripping while the alginate+PEDOT:PSS remains clogged.

Figure R4. Comparison of ink flow behavior in alginate-based PEDOT inks. Clogging observed in the printing nozzle for alginate+PEDOT:PSS inks, compared to the smooth flow of alginate+PEDOT:AlgS through the needle gauge. Air trapped in the syringe demonstrates the air pressure effect on the inks.

Changes: Supplementary Figure 15 was updated to include the images of printing fidelity:

Supplementary Figure 15 | Printability and SEM images of alginate-PEDOT composites. **a**, Clogging observed in the printing nozzle for alginate+PEDOT:PSS inks, compared to the smooth flow of alginate+PEDOT:AlgS through the needle gauge. Air trapped in the syringe demonstrates the air pressure effect on the inks. **b,c**, SEM images of alginate-PEDOT hydrogels containing **b**, 4% w/v PEDOT:PSS and **c**, 20% w/v PEDOT:AlgS.

Figure R4 was cited in the following sentence in section “Hydrogel-based 3D printed soft bioelectrodes”:

Changes (p. 10): *“In terms of printing fidelity (Figure 4e), solutions of 4% w/v PEDOT:PSS in alginate experienced multiple clogging events due to aggregation and precipitation (Supplementary Figure 15a).”*

Comment 14. On page 10, it is written that “The potential amplitudes for T waves of PEDOT:AlgS were found to be ~30% greater than PEDOT:PSS”, but this is not visible in Figure 4h. They show similar amplitudes.

Response: Please note that we specifically mentioned *T waves* in the ECG data and not the S waves. According to our results, while the S waves show comparable amplitudes between PEDOT:PSS and PEDOT:AlgS, the T waves in PEDOT:PSS group ranged between -0.07 and +0.09 mV (amplitude = 0.16 mV) and the T waves in PEDOT:AlgS ranged between -0.08 and +0.13 mV (amplitude = 0.21 mV). Thus, the amplitude of T waves in PEDOT:AlgS was 31.2% larger than PEDOT:PSS. We have revised the manuscript for further clarification.

Changes (p. 10): The following sentence was revised in section “Hydrogel-based 3D printed soft bioelectrodes”:

“While the potential amplitudes for S waves were found to be similar, these amplitudes for T waves were found to be approximately 30% greater than those observed for PEDOT:PSS.”

Comment 15. The control used for cell tests shown in Supplementary Figure 16 needs to be specified.

Response: We thank the Reviewer for bringing this issue to our attention. Blank controls are now specified in the figure caption.

Changes: The following clarification was added to the figure caption of Supplementary Figure 16: “Blank samples were used as controls.”

Comment 16. What the authors observed in Supplementary Figure 17 is not a sol-gel transition with the temperature, but the exact opposite, they observed a gel-sol transition. For example, at temperatures from 10 °C to 29 °C for GelCA, G' is higher than G'' , which is the condition for the gel state; however, from 29 °C, G'' is higher than G' , which is the conditions for the solution (sol) state. Therefore, the main text and the caption of this figure must be corrected accordingly.

Response: We thank the Reviewer for their attention to this important point. We have revised the labels in the figure, figure caption, as well as the main text to reflect the gel-sol transition in the results of Supplementary Figure 17.

Changes (p. 11): The following sentence was edited in section “Injectable smart bioadhesives for wound monitoring”:

“We observed an increase in viscosity (Figure 5c) and gel-sol transition temperature (Supplementary Figure 17) of GelCA with the addition of 20% w/v PEDOT:AlgS, comparable to 4% w/v PEDOT:PSS in GelCA.”

Supplementary Figure 17 and its caption were revised as follows:

Supplementary Figure 17 | Effect of PEDOT additives on thermal gel-sol transition of GelCA bioadhesive hydrogels. The gel-sol transition point is designated as the crossover of storage (G') and loss (G'') modulus.

Comment 17. How did the authors determine the tensile strain at failure (Supplementary Figure 18d)? I suppose those values were obtained from stress-strain curves shown in Supplementary Figure 18a, and I do not understand why there is no difference between GelCA+PEDOT:PSS and GelCA+PEDOT:AlgS for tensile at failure when there are large differences in the curves of the Supplementary Figure 18a.

Response: We defined the failure point as where the first drop was seen in the stress-strain curves and at that point (not where stress reaches back to zero). At this failure point, stress is registered as the tensile strength and strain is recorded as the tensile strain at failure. In essence, the failure point is attributed to where the crack propagates through the hydrogel cross-section for the first time and did not take into account deformations while crack propagation into the tensile strain at failure. With this definition, the non-significant differences in Supplementary Figure 18a are in accord with the stress-strain curves represented in Supplementary Figure 18a. We added further details on our definition of failure point in the methods section.

Changes (p. 19): The following sentence was added in section “Mechanical tensile tests”:

“The failure point was defined as the first drop observed in the stress-strain curves, from which the tensile strength and strain at failure were determined.”

Comment 18. The results of the clotting time corresponding to the blank control should be included in the bar plot in Figure 5f.

Response: The results of the decrease in clotting time shown in bar plots of Figure 5f are obtained relative to the control and thus, including the blank control would not be meaningful. We have included the absolute value of clotting time in addition to the representation of blood clotting time assay in this figure for more clarity.

Changes (p. 37): The figure caption of Figure 5f was improved as follows:

“Clotting time assays in terms of relative decrease in coagulation time for the assessment of hemostatic activity after hydrogel crosslinking. Clotting time for blank controls was 21.7 ± 0.6 min.”

Comment 19. What are the differences between the two bars of the plot included in the Supplementary Figure 22e? Both bars show the same sample name.

Response: We greatly appreciate the Reviewer for catching this error in Supplementary Figure 22e. The right bar in the bar plot belonged to GelCA+PEDOT:PSS group and is now corrected in the revised version.

Changes: Supplementary Figure 22e was corrected as shown below:

Supplementary Figure 22 | pH sensing capabilities of conductive bioadhesives. a,b, Real-time data of chronoamperometry tests showing the dependence of conductivity to different pH values

for GelCA (12% w/v) + PEDOT:PSS (4% w/v) and + PEDOT:AlgS (20% w/v), respectively. The tests were conducted in buffer solutions with varying pH levels (pH 4, 7, and 9) while applying a potential of 0.65 V. **c**, Open circuit potential results comparing pH sensitivity of bioadhesives. **d**, Current stability of the crosslinked GelCA+PEDOT:AlgS hydrogels at different pH levels. **e**, *In vivo* sensing of wound infection after 3 days post wound closure. CE, counter electrode; WE, working electrode; RE, reference electrode. The data represent mean \pm standard deviation (n=3).

Comment 20. On page 12, it is mentioned that “The pH sensing mechanism in GelCA-PEDOT is attributed to synergistic effects of mediated electron and ion mobility due to hydrogel swelling”. To prove this, swelling tests should be performed and the results should be included in the supporting information file.

Response: We truly appreciate the Reviewer for this comment. As per this suggestion, we conducted swelling experiments; however, we observed that while GelCA and GelCA-PEDOT:PSS swell significantly in water, GelCA-PEDOT:AlgS remains stable and does not absorb water possibly due to dense crosslinking as a result of high solid concentration (Figure R5).

Figure R5. Swelling ratio of GelCA and its hybrid combinations with PEDOT:PSS (4 %w/v) and PEDOT:AlgS (20 %w/v)

Thus, although this observation can support our assumed pH-sensitivity mechanism in GelCA-PEDOT:PSS, this assumption is not supported for GelCA-PEDOT:AlgS by swelling results due to the observed lack of swelling in water. Hence, we revised our statement and included the swelling data to avoid confusion.

Changes (p. 12): The following statement was added to section “Injectable smart bioadhesives for wound monitoring”:

“Results of swelling tests suggested substantial swelling in the GelCA and GelCA+PEDOT:PSS groups while no swelling was observed in the GelCA+PEDOT:AlgS groups due to the larger content of hydrophobic moieties and denser ionic crosslinking enabled by AlgS (Supplementary Figure 20b).”

The results of swelling tests were added to Supplementary Figure 20b:

Supplementary Figure 20 | Assessment of wet stability of crosslinking and static adhesion of conductive bioadhesives to collagen sheets. (a) Static adhesion of bioadhesive GelCA loaded with PEDOT:PSS and PEDOT:AlgS at their dispersibility limits after incubation for 24 h at 37 °C. (b) Swelling kinetic of crosslinked hydrogels incubated in water.

Changes (p. 12): The sentence below was revised in section “Injectable smart bioadhesives for wound monitoring”:

“The pH sensing in conductive hydrogels is generally attributed to synergistic effects of mediated electron and ion mobility, facilitated by hydrogel swelling, as well as catechol-to-quinone transitions induced by oxidation reactions^{54,60,61}. However, we note that swelling effect was found to be negligible in GelCA-PEDOT:AlgS, whereas it was more prominent under other conditions (Supplementary Figure 20b).”

A section “swelling tests” was added to Methods:

“**Swelling tests.** Water absorption of hydrogels was assessed via swelling experiments. Hydrogels were crosslinked and their initial wet weight was recorded. They were then soaked in DI water at room temperature and their weight was registered at different time points. The swelling ratio was calculated as the ratio of weight change to initial weight.”

Comment 21. The caption of Figure 5 needs to be revised as the label (E) should be replaced by e.

Response: Thank you for pointing that out. The label title has been now replaced by lower case e in the revised manuscript.

References

- R1. Sakunpongpitorn P, Phasuksom K, Paradee N, Sirivat A. Facile synthesis of highly conductive PEDOT:PSS *via* surfactant templates. *RSC Adv.* **9**, 6363-6378 (2019).

- R2. Cho H, Cho W, Kim Y, Lee JG, Kim JH. Influence of residual sodium ions on the structure and properties of poly(3,4-ethylenedioxythiophene):poly(styrenesulfonate). *RSC Adv.* **8**, 29044-29050 (2018).
- R3. Gorroñogoitia I, Urtaza U, Zubiarrain-Laserna A, Alonso-Varona A, Zaldua AM. A study of the printability of alginate-based bioinks by 3D bioprinting for articular cartilage tissue engineering. *Polymers* **14**, 354 (2022).
- R4. Volpatti LR, Bochenek MA, Facklam AL, Burns DM, MacIsaac C, Morgart A, *et al.* Partially oxidized alginate as a biodegradable carrier for glucose-responsive insulin delivery and islet cell replacement therapy. *Adv. Healthcare Mater.* **12**, 2201822 (2023).